# GRAIL: Graph Edit Distance and Node Alignment using LLM-Generated Code

**Samidha Verma** [1] * **Arushi Goyal** [2] * **Ananya Mathur** [2] * **Ankit Anand** [3] **Sayan Ranu** [1] [2]

## Abstract

Graph Edit Distance (GED) is a widely used metric for measuring similarity between two graphs. Computing the optimal GED is NP-hard, leading to the development of various neural and non-neural heuristics. While neural methods have achieved improved approximation quality compared to non-neural approaches, they face significant challenges: *(1)* They require large amounts of ground truth data, which is itself NP-hard to compute. *(2)* They operate as black boxes, offering limited interpretability. *(3)* They lack cross-domain generalization, necessitating expensive retraining for each new dataset. We address these limitations with GRAIL, introducing a paradigm shift in this domain. Instead of training a neural model to predict GED, GRAIL employs a novel combination of large language models (LLMs) and automated prompt tuning to generate a *program* that is used to compute GED. This shift from predicting GED to generating programs imparts various advantages, including end-to-end interpretability and an autonomous self-evolutionary learning mechanism without ground-truth supervision. Extensive experiments on seven datasets confirm that GRAIL not only surpasses state-of-the-art GED approximation methods in prediction quality but also achieves robust cross-domain generalization across diverse graph distributions.

## 1. Introduction and Related Work

*Graph Edit Distance* (GED) quantifies the dissimilarity between two graphs as the minimum number of *edits* required to transform one graph into another. An edit may comprise adding or deleting nodes and edges or replacing node

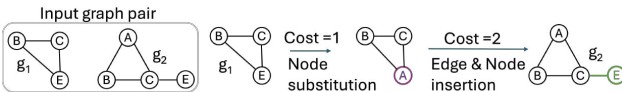

**Figure 1.** Illustration of edit path from $g_1$ to $g_2$ with GED 3.

and edge labels. Fig. 1 presents an example. Computing GED is NP-hard (Ranjan et al., 2022) and APX-hard (Fan et al., 2020; Bommakanti et al., 2024a). Owing to its numerous applications (Blumenthal, 2019; Ranu et al., 2014; Bommakanti et al., 2024b), polynomial-time heuristics are designed in practice.

### 1.1. Existing Works and their Limitations

Existing heuristics to approximate GED can be broadly grouped into two paradigms: non-neural and neural.

**Non-Neural Methods:** Computing GED exactly or approximating it within provable bounds is challenging, leading to the development of various heuristic approaches (Blumenthal et al., 2020). These methods utilize techniques such as transformations to the linear sum assignment (NODE (Justice & Hero, 2006), BRANCH-TIGHT (Blumenthal & Gamper, 2018)), mixed integer programming (MIP) (LP-GED-F2 (Lerouge et al., 2017a), ADJ-IP (Justice & Hero, 2006), COMPACT-MIP (Blumenthal & Gamper, 2020)), and local search methods (IPFP (Leordeanu et al., 2009)).

Unlike black-box neural methods, these approaches not only approximate GED but also provide the edit path, offering insights into structural modifications. However, their approximation quality is often inferior to neural approaches (Ranjan et al., 2022; Bai et al., 2019), driving the shift toward neural architectures. Additionally, these methods often involve solving complex optimization problems, such as MIP, to derive node alignments between graphs.

**Neural Methods:** Recent advancements favor graph neural networks (GNNs) for GED approximation due to their superior accuracy over non-neural methods (Ranjan et al., 2022; Zhang et al., 2021; Bai et al., 2019; Piao et al., 2023; Wang et al., 2021; Zhuo & Tan, 2022; Jain et al., 2024; Bai et al., 2020; Doan et al., 2021; Li et al., 2019). These models take pairs of graphs with known GED values as input and are trained to predict GED distances. However, since computing true GED is NP-hard, training these models efficiently for large graphs or datasets remains a significant challenge.

---

*Equal contribution [1]Yardi School of Artificial Intelligence, IIT Delhi, India [2]Department of Computer Science and Engineering, IIT Delhi, India [3]Google DeepMind, Montreal, Canada. Correspondence to: Samidha Verma < samidha.verma@scai.iitd.ac.in>, Arushi Goyal <cs5200418@iitd.ac.in>, Ananya Mathur <cs5200416@iitd.ac.in>, Ankit Anand <anandank@google.com>, Sayan Ranu <sayanranu@iitd.ac.in>.

*Proceedings of the $42^{nd}$ International Conference on Machine Learning*, Vancouver, Canada. PMLR 267, 2025. Copyright 2025 by the author(s).

| Name | End-to-end interpretable | Cross-domain generalization | Non-reliant on NP-hard supervision | Accurate |
|---|---|---|---|---|
| GREED (Ranjan et al., 2022) | X | X | X | ✔ |
| GEDGNN (Piao et al., 2023) | ○ | X | X | ✔ |
| H²MN (Zhang et al., 2021) | X | X | X | ✔ |
| ERIC (Zhuo & Tan, 2022) | X | X | X | ✔ |
| GRAPHEDX (Jain et al., 2024) | X | X | X | ✔ |
| GRAPHOTSIM (Doan et al., 2021) | X | X | X | ✔ |
| GRAPHSIM (Bai et al., 2020) | X | X | X | ✔ |
| TAGSIM (Bai & Zhao, 2021) | X | X | X | ✔ |
| GMN (Li et al., 2019) | X | X | X | ✔ |
| GENN-A* (Wang et al., 2021) | ○ | X | X | ✔ |
| SIMGNN (Bai et al., 2019) | X | X | X | ✔ |
| Non-neural approaches (Blumenthal et al., 2020) | ○ | ✔ | ✔ | X |
| GRAIL | ✔ | ✔ | ✔ | ✔ |

*Table 1.* Summary of the drawbacks of existing algorithms and the proposed algorithm GRAIL. ✔ indicates satisfaction of a desirable property, X indicates non-satisfaction, and ○ indicates partial satisfaction. While GEDGNN, GENN-A*, and traditional non-neural approaches achieve partial interpretability by providing edit paths corresponding to the GED, they do not explain the semantic reasoning behind these paths. In contrast, GRAIL achieves end-to-end interpretability through its code-based output, where each decision can be traced to its underlying logical reasoning. Non-neural approaches utilize unsupervised learning, enabling cross-domain generalization. However, their approximation errors are significantly higher on average than neural approaches, as demonstrated in § 5.

Among the leading algorithms, GREED (Ranjan et al., 2022) employs siamese GNNs with an inductive bias to learn GED while preserving its metric properties. H²MN (Zhang et al., 2021) utilizes a hierarchical hypergraph matching network for graph similarity learning. Other state-of-the-art approaches, such as (Piao et al., 2023), ERIC (Zhuo & Tan, 2022), and GraphEdX (Jain et al., 2024), further explore GNN-based architectures for GED prediction. Despite superior approximation quality, these methods suffer from key limitations given below.

- **Lack of interpretability:** Most neural methods only predict the GED and not the corresponding edit path. The edit path is essential for various applications such as identifying functions of protein complexes (Singh et al., 2008), image alignment (Conte et al., 2003), and uncovering gene-drug regulatory pathways (Chen et al., 2019). Few neural methods that predict the edit path (Piao et al., 2023; Wang et al., 2021) rely on expensive ground truth computation, which can only be attained for very small graphs (≈ 10 nodes). For larger graphs, random edits are made to synthetic graphs to generate the training samples.

- **NP-hard training data:** The training dataset for neural methods consists of graph pairs and their true GED. GED computation is NP-hard. Therefore, generating this training data is prohibitively expensive and restricted to small graphs only. Hence, approximation error deteriorates on larger graphs. (Ranjan et al., 2022)

- **Lack of generalization:** Neural GED approximators struggle to generalize across datasets. For datasets from different domains (e.g., chemical compounds vs. function-call graphs), the node label sets often differ. Since the number of parameters in a GNN depends on the feature dimensions of the nodes, GNNs fail to generalize across domains. Even within the same domain, as demonstrated later in §5, distribution shifts in structural and node label

distributions lead to increased approximation error. This limitation necessitates generating ground-truth data and training separate models for each dataset. Given that generating training data is NP-hard, this pipeline becomes highly resource-intensive.

### 1.2. Contributions

We address the above-outlined limitations through GRAIL: Graph Edit Distance and Node Alignment using LLM-Generated Code. GRAIL introduces a paradigm shift in the domain of GED approximations through the following novel innovations.

- **Problem formulation:** We shift the learning objective from approximating GED to learning a *program* that approximates GED. This reformulation provides end-to-end interpretability, as each algorithmic decision can be traced to its underlying logical reasoning. Moreover, by elevating the output to a higher level of abstraction through code generation, we achieve superior generalization across datasets, domains, graph sizes, and label distributions.

- **LLM-guided program discovery:** The algorithmic framework of GRAIL is grounded on three novel design choices. First, we map the problem of approximating GED to *maximum weight bipartite matching*, where the weights of the bipartite graph are computed using an LLM-generated program. Second, the prompt provided to the LLM is tuned through an evolutionary algorithm (Romera-Paredes et al., 2024). Third, our prompt-tuning methodology eliminates the need for ground-truth GED data by designing a prediction framework where the prediction is guaranteed to be an *upper bound* to the true GED. Hence, minimizing the upper-bound is equivalent to minimizing the approximation error, thereby overcoming a critical bottleneck of existing neural approaches.

- **Comprehensive Empirical Evaluation:** Through extensive experiments across 6 datasets, we demonstrate

that GRAIL discovers *foundational* code-based heuristics. Specifically, these heuristics not only surpass the state-of-the-art methods in GED computation but also exhibit generalization across diverse datasets and domains. This crucial feature eliminates the need for costly dataset-specific training, thereby addressing a significant limitation of existing neural algorithms.

## 2. Preliminaries and Problem Formulation

**Definition 1** (Graph). *We represent a node-labeled undirected graph as $\mathcal{G}(\mathcal{V}, \mathcal{E}, \mathcal{L})$ where $\mathcal{V} = \{v_1, \cdots, v_{|\mathcal{V}|}\}$ is the set of nodes, $\mathcal{E} \subseteq \mathcal{V} \times \mathcal{V}$ is the edge set and $\mathcal{L} : \mathcal{V} \to \Sigma$ is a labeling function that maps nodes to labels, where $\Sigma$ is the set of all labels.*

In unlabeled graphs, all nodes are assigned the same label.

**Definition 2** (Node Mapping). *A node mapping between two graphs $\mathcal{G}_1$ and $\mathcal{G}_2$, each consisting of $n$ nodes, refers to a bijection $\pi : \mathcal{V}_1 \to \mathcal{V}_2$, where every node $v \in \mathcal{V}_1$ is uniquely mapped to a node $\pi(v) \in \mathcal{V}_2$.*

**Extension to graphs of different sizes:** When dealing with two graphs $\mathcal{G}_1$ and $\mathcal{G}_2$ with different numbers of nodes, $n_1$ and $n_2$ respectively, such that $n_1 < n_2$, the smaller graph $\mathcal{G}_1$ can be extended to match the size of $\mathcal{G}_2$ by introducing $n_2 - n_1$ additional isolated *dummy* nodes. These new nodes are labeled with a unique identifier, $\epsilon$, indicating that they are placeholders with no connections. From this point onward, we assume that any pair of graphs in consideration have an equal number of nodes, with smaller graphs being augmented by dummy nodes as necessary.

**Definition 3** (GED under a node mapping $\pi$). *Given a node mapping $\pi$, the cost function for calculating graph edit distance between graphs $\mathcal{G}_1(\mathcal{V}_1, \mathcal{E}_1, \mathcal{L}_1)$ and $\mathcal{G}_2(\mathcal{V}_2, \mathcal{E}_2, \mathcal{L}_2)$ is expressed as:*

$$\text{GED}_\pi(\mathcal{G}_1, \mathcal{G}_2) = \sum_{v_1 \in \mathcal{V}_1} \mathbb{I}(\mathcal{L}_1(v_1) \neq \mathcal{L}_2(\pi(v_1)))$$
$$+ \frac{1}{2} \sum_{v_1 \in \mathcal{V}_1} \sum_{v_2 \in \mathcal{V}_1} \mathbb{I}(e_1(v_1, v_2) \neq e_2(\pi(v_1), \pi(v_2)))$$

*where,*
- *$e_i(u, v)$ returns 1 if the edge $(u, v) \in \mathcal{E}_i$ in graph $\mathcal{G}_i$, 0 otherwise.*
- *$\mathbb{I}(A)$ is the indicator function, which is 1 if the condition $A$ holds, and 0 otherwise.*

**Interpretation of edit path from node mapping:** The first part of the equation captures node mismatches where it evaluates the label differences between nodes in $\mathcal{G}_1$ and $\mathcal{G}_2$. Mapping a dummy node to a real node (or vice versa) results in a label mismatch, reflecting the insertion or deletion of a node, while a mismatch between real nodes denotes a substitution. The second part of the equation captures edge mismatches. Specifically, if an existing edge in $\mathcal{G}_1$ (i.e., $e_1(v_1, v_2) = 1$) is mapped to a non-existing edge in $\mathcal{G}_2$ (i.e., $e_2(\pi(v_1), \pi(v_2)) = 0$) or vice versa, the cost is 1 representing edge deletion and insertion, respectively.

**Definition 4** (Graph edit distance (GED)). *The GED between graphs $\mathcal{G}_1$ and $\mathcal{G}_2$ is the minimum GED across all possible node mappings.*

$$\text{GED}(\mathcal{G}_1, \mathcal{G}_2) = \min_{\forall \pi \in \mathcal{M}} \{\text{GED}_\pi(\mathcal{G}_1, \mathcal{G}_2)\} \tag{1}$$

*Here, $\mathcal{M}$ denotes the universe of all possible mappings.*

The problem is hard (NP-hard and APX-hard) since the cardinality of $\mathcal{M}$ is $n!$, where $n = \max\{|\mathcal{V}_1|, |\mathcal{V}_2|\}$.

The problem of learning to code for approximating GED is defined as follows.

**Problem 1** (Learning to code for GED). *Given a set of training graph pairs $\mathbb{T} = \{\langle \mathcal{G}_1, \mathcal{G}_1' \rangle, \langle \mathcal{G}_2, \mathcal{G}_2' \rangle, \cdots, \langle \mathcal{G}_n, \mathcal{G}_n' \rangle\}$, learn a program $P : (\mathcal{G}_t, \mathcal{G}_t') \to \mathbb{Z}+$ that takes as input a graph pair $\langle \mathcal{G}_t, \mathcal{G}_t' \rangle \in \mathbb{T}$, and outputs a non-negative integral distance that minimizes*

$$\sum_{t=1}^{n} |P(\mathcal{G}_t, \mathcal{G}_t') - \text{GED}(\mathcal{G}_t, \mathcal{G}_t')| \tag{2}$$

Note that our training set consists solely of graph pairs, without requiring their true GED, which is computationally prohibitive due to its NP-hardness. As we will elaborate in the next section, we identify polynomial-time computable upper bounds for the true GED and reformulate the optimization objective to minimize this upper bound. This autonomous self-evolutionary learning mechanism overcomes a significant limitation of neural GED approximators.

## 3. Approximation Strategy

The true GED corresponds to the minimum distance across all possible node mappings (Def. 4). However, enumerating all such mappings is computationally infeasible due to its factorial complexity relative to graph size. To overcome this challenge, we approximate the GED by evaluating a small subset of mappings (e.g., 15) and selecting the minimum distance among them. These mappings are generated by programs derived from the LLM, as detailed in § 4. Importantly, this approximated GED serves as an upper bound to the true GED, as it considers only a subset of all possible node mappings.

### 3.1. Node Mappings through Bipartite Matching

The task of mapping nodes between two graphs can be approximated as *Maximum Weight Bipartite Matching*.

**Definition 5** (Maximum Weight Bipartite Matching). *Given a weighted bipartite graph $\mathcal{B}(\mathcal{V}, \mathcal{U}, \mathcal{E}, \mathcal{W})$ with node sets $\mathcal{V}$ and $\mathcal{U}$, and a weighted edge set $\mathcal{E} : \mathcal{V} \times \mathcal{U} \to \mathbb{R}$ where $\mathcal{W} : \mathcal{E} \to \mathbb{R}$ assigns weights to edges, find a subset of edges $\mathcal{E}^* \subseteq \mathcal{E}$ that (1) induces a bijection between nodes in $\mathcal{V}$ and nodes in $\mathcal{U}$, and (2) maximizes the total weight of the mapped edges, i.e., $\sum_{e \in \mathcal{E}^*} \mathcal{W}(e)$. Here, $\mathcal{W}(e)$ represents the weight of edge $e$.*

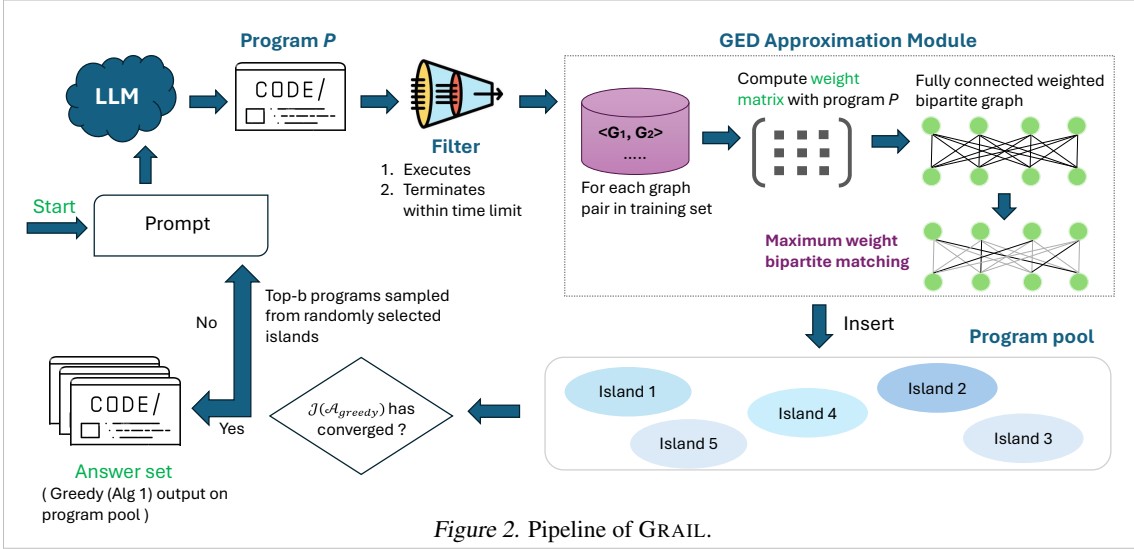

Figure 2. Pipeline of GRAIL.

Maximum weight bipartite matching can be solved optimally in polynomial time using the Hungarian algorithm (Kuhn, 1955). Additionally, several heuristics have been proposed, such as the Hopcroft–Karp Algorithm (Hopcroft & Karp, 1973), the Neighbor-biased mapper (He & Singh, 2006), or greedy selection of the highest-weight edges while maintaining the one-to-one mapping constraint. We use the notation $\pi(\mathcal{B})$ to denote the node mapping obtained from $\mathcal{B}$.

To use maximum weight bipartite matching for approximating GED, for given graphs $\mathcal{G}_1(\mathcal{V}_1, \mathcal{E}_1, \mathcal{L}_1)$ and $\mathcal{G}_2(\mathcal{V}_2, \mathcal{E}_2, \mathcal{L}_2)$, we construct a fully connected, weighted bipartite graph $\mathcal{B}(\mathcal{V}_1, \mathcal{V}_2, \mathcal{E}, \mathcal{W})$ where edge set $\mathcal{E} = \{(v_1, v_2) \mid v_1 \in \mathcal{V}_1, v_2 \in \mathcal{V}_2\}$[1]. The weight of an edge $(v_1, v_2)$ is set based on some policy, which should ideally reflect the probability of $v_1$ being mapped to $v_2$ in the optimal GED mapping. Maximum weight bipartite matching is then performed on $\mathcal{B}$ using any standard algorithm, and the GED is computed based on the mapping $\pi(\mathcal{B})$.

The quality of the mapping with respect to approximating GED, therefore, rests on the edge weights in the bipartite graph. We will use an LLM to learn the policy, in the form of a program, with the following *minimization objective*.

**Problem 2** (Weight Matrix Generation). *Given train set* $\mathbb{T} = \{\langle \mathcal{G}_1, \mathcal{G}_1' \rangle, \cdots, \langle \mathcal{G}_n, \mathcal{G}_n' \rangle\}$, *generate a program* $P$ *that takes as input each pair* $\langle \mathcal{G}_t, \mathcal{G}_t' \rangle \in \mathbb{T}$ *and outputs a corresponding weight matrix* $\mathbf{W}_t \in \mathbb{R}^{|\mathcal{V}_t| \times |\mathcal{V}_t'|}$ *minimizing*
$$\sum_{t=1}^{n} \text{GED}_{\pi(P)}(\mathcal{G}_t, \mathcal{G}_t') \qquad (3)$$
*Here,* $\mathbf{W}_t[i, j]$ *denotes the weight of edge* $(v_i, v_j)$ *where* $v_i \in \mathcal{V}_t$ *to node* $v_j \in \mathcal{V}_t'$. $\pi(P)$ *denotes the mapping generated by maximum weight bipartite matching on the bipartite graph formed by* $P$.

[1] Recall, we assume the smaller graph is padded with dummy nodes to ensure $|\mathcal{V}_1| = |\mathcal{V}_2|$.

### 3.2. Budget-constrained Selection of Node Maps

Let $\mathcal{D} = \{P_1, \cdots, P_m\}$ be the set of programs generated by the LLM and $\pi(P_i)$ denote the mapping produced by program $P_i \in \mathcal{D}$. From Def. 4, we know $\text{GED}_{\pi(P_i)}(\mathcal{G}_1, \mathcal{G}_2) \geq \text{GED}(\mathcal{G}_1, \mathcal{G}_2)$, i.e., each program provides an *upper bound* on the true GED. The smaller the upper bound, the closer we are to the true GED. Our goal is to select a subset $\mathcal{A}^* \subseteq \mathcal{D}$ of $b$ programs that minimize the cumulative upper bounds across all train graph pairs. $b \ll |\mathcal{D}|$ denotes the maximum number of mappings we are allowed to evaluate. These $b$ programs will finally be used during inference for unseen graph pairs. Formally, this presents us with the following optimization problem.

**Problem 3** (Map Selection). *Given programs* $\mathcal{D} = \{P_1, \cdots, P_m\}$, *select* $\mathcal{A}^*$ *such that:*
$$\mathcal{A}^* = \arg \min_{\forall \mathcal{A} \subseteq \mathcal{D}, |\mathcal{A}| = b} \{\mathcal{J}(A)\} \qquad (4)$$
$$\mathcal{J}(\mathcal{A}) = \sum_{\langle \mathcal{G}_1, \mathcal{G}_2 \rangle \in \mathbb{T}} \min_{P \in \mathcal{A}} \{\text{GED}_{\pi(P)}(\mathcal{G}_1, \mathcal{G}_2)\} \qquad (5)$$

$\mathcal{J}(\mathcal{A})$ *quantifies the quality of the subset of mappings in* $\mathcal{A}$.

**Theorem 3.1.** *Prob. 3 is NP-hard.*

*Proof.* The proposed optimization problem reduces to the *Set Cover* problem (Cormen et al., 2009), rendering it NP-hard. For the formal proof, please refer to App. A.2.1. □

Owing to NP-hardness, finding the optimal subset of mappings $\mathcal{A}^*$ is not feasible in polynomial time. We establish that $\mathcal{J}(\mathcal{A})$ is *monotonic* and *submodular* (refer App. A.2.2). This enables us to use the greedy hill-climbing algorithm (Alg. 1) to select a sub-optimal but reasonable subset of programs, $\mathcal{A}_{greedy}$.

## 4. GRAIL: Proposed Methodology

In § 6, we decompose Prob. 1 into two subproblems: weight selection in a bipartite graph (Prob. 2) and budget-

constrained map selection (Prob. 3). Prob. 3 is solved (approximately) using Alg. 1. Hence, to complete our approximation scheme, we need to solve Prob. 2.

Fig. 2 presents the pipeline of GRAIL. The process begins with an initial prompt that specifies a trivial program for weight selection, and the LLM is tasked with improving this program for GED computation via bipartite matching (details in § 4.1). Each newly generated program is verified for syntactic correctness and must terminate within a predefined time limit. If these criteria are met, the program is evaluated on the training set of graph pairs and added to the program pool along with its *score*, which reflects its marginal contribution to $\mathcal{J}(\mathcal{A}_{greedy})$. A new prompt is then constructed by sampling the highest-scoring programs from the current pool. The LLM refines these programs, generating new candidates to further enhance performance. These newly generated programs are evaluated and added to the pool following the same procedure. This iterative process continues until $\mathcal{J}(\mathcal{A}_{greedy})$ converges, ensuring that improvements stabilize across iterations. The following sections detail each step of this process.

### 4.1. Prompt Specification

The prompt is a computer program consisting of three distinct components: **(1)** the problem description, **(2)** the task specification, and **(3)** the top-$k$ programs generated so far based on a scoring function. A sample prompt is provided in Fig. 4 in the Appendix. Here, $k$ is set to 2.

**Problem Description:** The problem description includes the definition of GED, which is embedded as a comment within the program (refer to Fig. 4).

**Task Specification:** The LLM's task is defined through a comment specifying the inputs it should expect and the required output. The output is a weight matrix $\mathbf{W} \in \mathbb{R}^{|\mathcal{V}_1| \times |\mathcal{V}_2|}$ for the bipartite graph, where $\mathbf{W}[i,j]$ quantifies the strength of mapping node $v_i \in \mathcal{V}_1$ to node $v_j \in \mathcal{V}_2$ in the context of GED computation. The input includes the graph pair represented by their adjacency matrices and an initial weight matrix $\mathbf{W}^0 \in \mathbb{R}^{|\mathcal{V}_1| \times |\mathcal{V}_2|}$ with the same dimensions and semantics as the output. During execution, the input weight matrix $\mathbf{W}^0$ is initialized such that $\mathbf{W}^0[i,j] = 1$ if

the corresponding nodes share the same label, i.e., $\mathcal{L}_1(v_i) = \mathcal{L}_2(v_j)$ for $v_i \in \mathcal{V}_1$ and $v_j \in \mathcal{V}_2$, and $\mathbf{W}^0[i,j] = 0$ otherwise. Additionally, the header of the function that the LLM needs to generate is explicitly provided.

**Top-$k$ Programs:** The initial prompt includes a trivial program where $\forall v_i \in \mathcal{V}_1, v_j \in \mathcal{V}_2, \mathbf{W}[i,j] = 0$. In subsequent iterations, $k$ high-scoring programs are sampled for inclusion in the prompt, where $k$ is a hyper-parameter. The scoring and sampling methodology are described in § 4.2.

### 4.2. Prompt Tuning

**Filter:** After a program is generated, it undergoes a filtering step to verify that it executes and terminates on training graph pairs within a predefined time limit. Programs that fail this filter are discarded. For those that pass, we compute their score and add them to our program database $\mathcal{D}$.

**Score computation:** In Prob. 3, we take the minimum GED across all selected mappings in the answer set. A program's utility, therefore, depends on how it complements other programs in the answer set. Hence, we define its score as the *marginal contribution* to the objective function $\mathcal{J}(\mathcal{A}_{\text{greedy}})$. Specifically, we execute Alg. 1 on the current pool of programs, where $\mathcal{A} = \{P_1, \ldots, P_i\}$ represents the subset of programs selected up to iteration $i$. If program $P$ is added in the $i+1$-th iteration due to providing the highest marginal contribution, its score is computed as:

$$\text{score}(P) = \mathcal{J}(\mathcal{A}) - \mathcal{J}(\mathcal{A} \cup \{P\}), \quad (6)$$

**Evolutionary program selection:** The next stage involves selecting programs from the pool to be included in the next prompt. We use the evolutionary algorithm proposed in Funsearch (Romera-Paredes et al., 2024) for evolving our programs generated by LLM. Since the programs evolve through mutations introduced by the LLM, the selection mechanism optimizes two distinct objectives. First, the sampled programs should have high scores. Second, the sampled programs should have smaller length improving the interpretability of generated programs.

The evolutionary algorithm follows the *islands model* (Gordon & Whitley, 1993). Specifically, the population of existing programs is partitioned into $s$ islands, where $s$ is a hyperparameter. Initially, all islands are empty. When a program is added to the pool, it is randomly assigned to an island. Subsequently, to decide which $k$ programs are included in the prompt, we randomly choose an island. Similar to Funsearch, the programs within each island are then split into clusters depending on score. After selecting the island, clusters are selected based on softmax distribution on score. Within a clusters, the programs are selected based on length (smaller is better). Hence, the program selection mechanism for the next prompt favors higher scores and shorter lengths. More details of the process can be found in Romera-Paredes et al. (2024). The LLM is then tasked with

---

**Algorithm 1** The greedy approach

**Require:** Train data $\mathbb{T} = \{T_1, \cdots, T_n\}$ where $T_t = \langle \mathcal{G}_t, \mathcal{G}'_t \rangle$ is a pair of graphs, budget $b$.
**Ensure:** solution set $\mathcal{A}_{greedy}$, $|\mathcal{A}_{greedy}| = b$
1: $\mathcal{A}_{greedy} \leftarrow \emptyset$
2: **while** $size(\mathcal{A}_{greedy}) \leq b$ (within budget) **do**
3: $\quad P^* \leftarrow \arg\max_{P \in \mathcal{D} \setminus \mathcal{A}_{\text{greedy}}}$
$\quad\quad\quad \{\mathcal{J}(\mathcal{A}_{\text{greedy}}) - \mathcal{J}(\mathcal{A}_{\text{greedy}} \cup \{P\})\}$
4: $\quad \mathcal{A}_{greedy} \leftarrow \mathcal{A}_{greedy} \cup \{P^*\}$
5: **Return** $\mathcal{A}_{greedy}$

---

further improving these programs.

With this design, each island evolves independently. To enable cross-fertilization among islands, we periodically discard half of the islands which have the lowest score. The discarded islands are replaced by iterating over each of the surviving islands, and selecting its best program to seed the replacement population.

### 4.3. Training and Inference

**Training:** As illustrated in Fig. 2, each iteration involves generating a program, scoring it, and assigning it to an island before constructing and executing a new prompt. The quality of the program pool is measured by $\mathcal{J}(\mathcal{A}_{greedy})$, serving as an analog to a loss function in our framework. This iterative process continues until $\mathcal{J}(\mathcal{A}_{greedy})$ converges, defined as its improvement over the last $i$ iterations falling below a predefined threshold, akin to the *patience* parameter in neural model training.

Overall, GRAIL seeks to minimize the upper bound of GED. With this strategy, we bypass the need for ground-truth GED data, a key bottleneck in training neural approaches. This unique design is not feasible in neural pipelines since the prediction can err on either side of the true distance.

**Inference:** During inference, we directly return $\mathcal{J}(\mathcal{A}_{greedy})$ for the given input graph pair. Note that since the output of the training phase is executable code, inference is CPU-bound, enabling it to operate in low-resource environments.

## 5. Experiments

In this section, we benchmark GRAIL and establish that:

- **Approximation Error:** GRAIL achieves low approximation errors and consistently ranks among the top algorithms across all six datasets. Notably, unlike neural approximators, it achieves this performance without relying on extensive NP-hard ground-truth GED training data.

- **Foundational heuristics:** GRAIL breaks new ground by generating heuristics that generalize across diverse datasets, including those featuring unseen node labels and varying graph sizes. This exceptional adaptability sets GRAIL apart, as no existing neural GED approximators have demonstrated such versatility.

The codebase of GRAIL and the programs generated for the various datasets are available at `https://github.com/idea-iitd/Grail`.

### 5.1. Experiment Setup

Gemini-1.5 Pro has been used for all experiments. Further details of the software and hardware environments and hyper-parameters used for GRAIL are listed in App. A.3.1.
**Datasets:** Table 2 summarizes the datasets used in this study. A detailed description of the data semantics is included in App. A.4. While AIDS, Linux and IMDB are obtained from

*Table 2.* Datasets used for benchmarking GRAIL.

| Name | # Graphs | Avg $|V|$ | Avg $|E|$ | # labels | Domain |
|---|---|---|---|---|---|
| ogbg-molhiv | 39650 | 24 | 52 | 119 | Molecules |
| ogbg-molpcba | 436313 | 26 | 56 | 119 | Molecules |
| ogbg-code2 | 139468 | 37 | 72 | 97 | Software |
| AIDS | 700 | 9 | 9 | 29 | Molecules |
| Linux | 1000 | 8 | 7 | Unlabeled | Software |
| IMDB | 1500 | 13 | 65 | Unlabeled | Movies |
| ogbg-ppa | 39650 | 243.4 | 2226.1 | Unlabeled | Protein |

Morris et al. (2020), the other four datasets are made available by Hu et al. (2021).
**Benchmark Algorithms:** The recent baselines are listed in Table 1. From this set, we benchmark GRAIL against GREED (Ranjan et al., 2022), GEDGNN (Piao et al., 2023), ERIC (Zhuo & Tan, 2022), GRAPHEDX (Jain et al., 2024) and H²MN (Zhang et al., 2021). We omit SIMGNN, GRAPHOTSIM, GMN, GRAPHSIM, TAGSIM and GENN-A*, since they have been outperformed by the considered baselines of GREED, GRAPHEDX, GEDGNN and ERIC. Details of setup are provided in App. A.1.

Among non-neural baselines we include the best-performing heuristics from the benchmarking study in Blumenthal et al. (2020): namely, LP-GED-F2, COMPACT-MIP, ADJ-IP, BRANCH-TIGHT, NODE and IPFP.

**GRAIL-MIX** is a variant of GRAIL trained on a mixture of graph pairs from multiple datasets, while maintaining the same training set size as GRAIL. The programs discovered by GRAIL-MIX are used for inference across all datasets to assess whether a single training instance can generalize across domains, eliminating dataset-specific training.

**Train-Validation-Test Split:** To construct the test set for a particular dataset, we select 1000 graph pairs uniformly at random and compute their true GED. The procedure for computing the ground truth GED is discussed in App. A.4.1. The training and validation sets depend on the algorithm.

- **Neural Algorithms:** All neural approaches are trained on 10,000 graph pairs per dataset. This training time exceeds 15 days for certain datasets (see Fig. 6a).

- **GRAIL and GRAIL-MIX:** GRAIL is trained with only 1,000 graph pairs per dataset. As discussed already, GRAIL does not require ground-truth GED. In GRAIL-MIX, we choose 166 graph pairs from each of the datasets listed in Table 2 except ogbg-ppa. Both GRAIL and GRAIL-MIX do not use a validation set.

- **Non-Neural Baselines:** These unsupervised algorithms do not require any training or validation datasets.

**Metrics:** We employ two metrics: Root Mean Squared Error (*RMSE*) and Exact Match Ratio (*EMR*). *EMR* quantifies the proportion of test graph pairs for which the predicted GED exactly matches the true GED. (See App. A.4.2 for details.)

*Table 3.* **RMSE Comparison:** The top-3 lowest RMSEs per dataset are highlighted in green shades, with darker shades denoting better RMSE. An asterisk (*) marks the better value when additional decimal places resolve ties after rounding to two decimal places.

| Type | Methods | AIDS | Linux | IMDB | ogbg-molhiv | ogbg-code2 | ogbg-molpcba | Avg. Rank |
|---|---|---|---|---|---|---|---|---|
| **LLM** | GRAIL | 0.57 | 0.13 | 0.55 | 2.96* | 4.22 | 3.18 | 2 |
| | GRAIL-MIX | 0.64 | 0.11 | 0.53 | 2.96 | 4.10 | 3.40 | 2.17 |
| **Neural** | GREED | 0.61 | 0.41 | 4.8 | 3.02 | 5.52 | 2.48 | 3.5 |
| | GEDGNN | 0.92 | 0.29 | 4.43 | 1.75 | 16.68 | 4.58 | 5 |
| | ERIC | 1.08 | 0.30 | 42.44 | 3.56 | 17.55 | 2.79 | 6.5 |
| | H$^2$MN | 1.14 | 0.60 | 57.8 | 12.01 | 11.96 | 5.50 | 8.33 |
| | GRAPHEDX | 0.78 | 0.27 | 32.36 | 14.14 | 21.46 | 10.01 | 8.33 |
| **Non Neural** | ADJ-IP | 0.85 | 0.50 | 42.18 | 10.21 | 14.94 | 8.06 | 7.33 |
| | NODE | 2.71 | 1.24 | 61.03 | 4.97 | 8.34 | 4.94 | 8.17 |
| | LP-GED-F2 | 1.96 | 0.23 | 55.26 | 12.86 | 16.03 | 10.30 | 8.83 |
| | BRANCH | 3.31 | 2.45 | 7.36 | 9.86 | 12.64 | 11.31 | 9.33 |
| | COMPACT-MIP | 2.69 | 0.44 | 65.88 | 10.88 | 19.46 | 8.81 | 10 |
| | IPFP | 4.18 | 2.29 | 69.45 | 13.69 | 15.19 | 10.02 | 11.5 |

*Table 4.* **EMR Comparison**: The top-3 highest EMRs per dataset are highlighted in green shades, with darker shades denoting better EMRs. The EMR values for GRAIL are in App. (Table 9). We omit them here as GRAIL-MIX performs similarly across datasets. For a focused comparison, we only include the top-3 baselines from Table 3, since the remaining do not provide competitive performance. For ties after rounding to two decimals, an asterisk (*) marks the higher value. Values in (0.99, 1) are shown as $\approx 1$.

| Methods | AIDS | Linux | IMDB | ogbg-molhiv | ogbg-code2 | ogbg-molpcba | Avg. Rank |
|---|---|---|---|---|---|---|---|
| GRAIL-MIX | 0.80 | $\approx 1$ | $\approx 1$ | 0.20 | 0.12 | 0.12 | 1.83 |
| GREED | 0.58 | 0.79 | 0.17 | 0.23 | 0.09 | 0.21 | 2.17 |
| ERIC | 0.37 | 0.92 | 0.08 | 0.21 | 0.01 | 0.18 | 2.83 |
| GEDGNN | 0.35 | 0.85 | 0.07 | 0.57 | 0.01* | 0.09 | 3.17 |

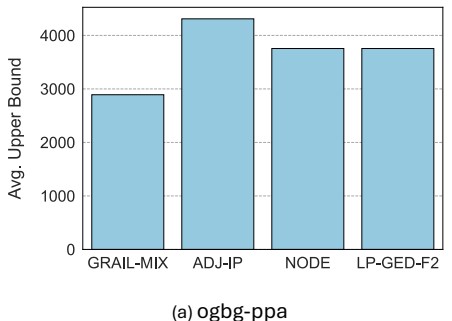

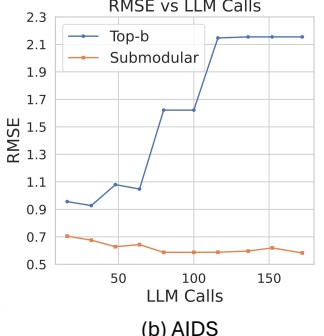

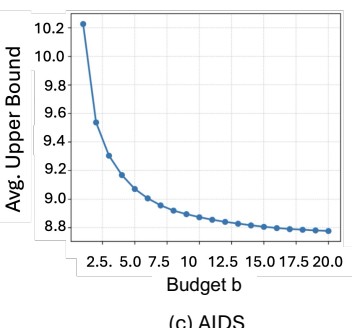

(a) ogbg-ppa   (b) AIDS   (c) AIDS

*Figure 3.* (a) **GRAIL-MIX at scale:** Performance of GRAIL-MIX on the ogbg-ppa dataset when compared to the top-3 non-neural baselines on the basis of average rank in Table 3. (b) Impact of function budget on upper bound. (c) Impact of greedy submodular optimization on performance on test set.

## 6. Approximation Strategy

### 6.1. Empirical Analysis of Approximation Errors

Tables 3 and 4 benchmark GRAIL in terms of RMSE and EMR. Several important observations emerge.

First, GRAIL and GRAIL-MIX comprehensively outperform the baselines despite not using any ground truth for training. This is a critical advantage as it saves time in expensive ground truth computation (Refer to Fig. 6a). The improvement is the highest in the IMDB dataset, which we specifically analyze in § 6.4

Second, the efficacy of GRAIL-MIX across datasets demonstrates that the discovered programs are universally applicable on multiple datasets and can be called as *foundation*

*functions*. These foundation functions eliminate the need for dataset-specific training.

Third, GRAIL-MIX outperforms GRAIL in three datasets, indicating positive cross-dataset knowledge transfer.

Fourth, we inspected why GRAIL significantly underperformed in comparison to GREED for ogbg-molpcba dataset. We observed that adapting the neighborhood depth used for computing node similarities can substantially improve performance. GREED leverages jumping knowledge to combine node embeddings from depths 1 to 7, allowing an MLP to learn which depth is most effective for GED approximation. In Tables 3 and 4 GRAIL uses a fixed topological depth across all nodes. When GRAIL adopts a similar strategy of computing GEDs across multiple depths and selecting the

*Table 5.* **Intra-Domain Generalizability:** RMSE of the best neural method, GREED, and GRAIL. Off-diagonal entries represent cross-dataset performance. NA symbolizes that it's not possible to train a single model covering the train-test combination.

| Test set \ Train Set | AIDS | | ogbg-molhiv | | ogbg-molpcba | |
|---|---|---|---|---|---|---|
| | GREED | GRAIL | GREED | GRAIL | GREED | GRAIL |
| **AIDS** | 0.61 | 0.57 | 5.71 | 0.64 | 4.58 | 0.59 |
| **ogbg-molhiv** | NA | 3.02 | 3.02 | 2.96 | 3.86 | 2.89 |
| **ogbg-molpcba** | NA | 3.59 | 2.16 | 3.54 | 2.48 | 3.18 |

*Table 6.* **Inter-Domain Generalizability:** Generalization of GRAIL across domains, dataset sizes, and node label distributions by training on one dataset and measuring RMSE on others. Off-diagonal entries represent cross-dataset performance. The best two results for each test set(row) have been highlighted in shades of green, with darker being better.

| Test set \ Train set | AIDS | IMDB | Linux | ogbg-molhiv | ogbg-code2 | ogbg-molpcba |
|---|---|---|---|---|---|---|
| **AIDS** | 0.57 | 0.63 | 0.65 | 0.64 | 0.62 | 0.59 |
| **IMDB** | 0.88 | 0.55 | 0.88 | 0.78 | 0.74 | 0.87 |
| **Linux** | 0.18 | 0.22 | 0.13 | 0.24 | 0.16 | 0.24 |
| **ogbg-molhiv** | 3.02 | 2.93 | 3.08 | 2.96 | 2.96 | 2.89 |
| **ogbg-code2** | 4.44 | 4.32 | 4.74 | 4.07 | 4.22 | 4.5 |
| **ogbg-molpcba** | 3.59 | 3.63 | 3.61 | 3.54 | 3.64 | 3.18 |

minimum as the final estimate, its RMSE improves from 3.18 (depth 1) to 2.68 (depths 1,2,3), becoming competitive with GREED 's RMSE of 2.48. This suggests that dynamically adapting neighborhood depth can further improve the approximation quality of GRAIL.

As an extended experiment, we compare our method with an ensemble of 15 non-neural methods from (Blumenthal et al., 2020) and show that GRAIL performs better in both RMSE and EMR metrics while being 168x faster. The detailed results are provided in App. A.5.1.

We compare the minimum, maximum, and average RMSE of individual functions discovered by GRAIL and non-neural heuristics in App. A.5.2. GRAIL achieves better average and maximum RMSE across all datasets, while the non-neural ensemble has better minimum RMSE in 4 out of 6 datasets. This is expected, as GRAIL focuses on learning diverse, complementary heuristics via a submodular loss, rather than optimizing for a single best function.

### 6.2. Generalizability

An intrinsic requirement of all machine learning methods is that the training and test data are sampled from the same distribution. Thus, the neural baselines depend on training data tailored to the test dataset, limiting their ability to transfer knowledge due to reliance on dataset-specific features. In contrast, GRAIL learns symbolic logical rules in the form of programs, facilitating out-of-domain and out-of-distribution generalization. We now evaluate this capability.

**Intra-domain:** Neural models are limited by feature dimensionality, making zero-shot generalization infeasible. However, for domains such as chemical compounds, a uniform feature space allows training a single model. We retrain the best neural baseline, GREED, and compare its intra-domain generalizability with GRAIL in Table 5. Note that the AIDS dataset has a smaller feature dimension than

the ogbg datasets, making it impossible to derive a common feature space. GRAIL generalizes well across all train-test dataset pairs, while GREED struggles except for ogbg-molhiv and ogbg-molpcba. This is because both datasets are adopted from the same parent dataset MoleculeNet (Wu et al., 2018; Hu et al., 2020) with similar topological features (Table 2).

**Inter-domain:** Table 6 showcases the ability of the functions discovered by dataset-specific training of GRAIL to generalize across other datasets. We do not observe a significant increase in RMSE on the off-diagonal entries, which showcases positive knowledge transfer and an ability not seen in neural approximators.

**Generalization to graph size:** In this experiment, we evaluate GRAIL-MIX on the ogbg-ppa dataset, which has been omitted from prior benchmarking due to the large size of its graphs (See Table 2). Given the computational infeasibility of ground-truth GED computation for these graphs, neural approximators cannot be trained on this dataset. To assess the generalization capability of GRAIL-MIX, we compare the upper bound provided by its programs against the top-3 non-neural heuristics (based on Table 3). The results are illustrated in Fig. 3a. We observe that GRAIL-MIX provides 30% to 45% tighter upper bounds. Fig. 7 in the appendix further substantiates that GRAIL generalizes better to large graphs than neural baselines.

### 6.3. Ablation Study and Parameters

GRAIL comprises of two main components: the GED *Approximation Module* that reduces GED approximation to bipartite matching and learns weights of the bipartite graph using an LLM (Refer § 3) and *Prompt-Tuning* via evolutionary program selection (Refer § 4.1). In this section, we conduct an ablation study to assess the utility of each component and the impact of the budget parameter.

*Table 7.* **Ablation Study:** Comparison of GRAIL with variants without GED Approximation Module ("Direct fns (15)") and Evolutionary Prompt Selection (Random). The best performing method has been highlighted in green.

| Metric | Method | AIDS | Linux | IMDB | ogbg-molhiv | ogbg-code2 | ogbg-molpcba |
|--------|--------|------|-------|------|-------------|------------|--------------|
| RMSE | GRAIL | 0.57 | 0.13 | 0.55 | 2.96 | 4.22 | 3.18 |
|  | Direct fns (15) | 7.95 | 2.81 | 112.64 | 7.47 | 25.68 | 9.71 |
|  | Random | 0.94 | 0.38 | 0.93 | 3.48 | 4.42 | 4.19 |
| EMR | GRAIL | 0.83 | $\approx 1$ | 0.99 | 0.18 | 0.11 | 0.12 |
|  | Direct fns (15) | 0.01 | 0.14 | 0.09 | 0.02 | 0 | 0.02 |
|  | Random | 0.67 | 0.97 | 0.97 | 0.14 | 0.10 | 0.07 |

**Utility of GED Approximation Module:** Here, we investigate whether the reduction to bipartite matching is necessary. Specifically, we directly prompt the LLM to generate code for computing GED, bypassing the intermediate step of predicting weights for a bipartite graph. To ensure a fair comparison, we adopt the same setup as GRAIL, using an ensemble of 15 heuristics and selecting the minimum predicted value across each graph pair as the final GED. As shown in Table 7, performance deteriorates significantly under this setting.

**Impact of Evolutionary Prompt Selection:** We assess the impact of the genetic evolution as described in § 4.1, by randomly selecting programs from the program pool for the prompt instead of using the genetic algorithm. The RMSE increases by 2 to 3 times on average, as shown in Table 7.

**Impact of Submodularity:** What happens if, instead of selecting the top-$b$ functions using greedy submodular optimization, we evolve and score functions individually based on their upper bounds (Eq. 3) and select the top-$b$ solely based on this criterion? Fig. 3b illustrates the impact on RMSE across training iterations (LLM calls). While selecting the top-$b$ functions through submodular optimization shows a clear trend of decreasing RMSE on the test set, independently choosing the top-$b$ functions based on individual scores results in significantly higher RMSE, with a progressive worsening trend indicative of overfitting to the training set. This result underscores the importance of submodularity in selecting functions that complement one another and perform well collectively (See Fig. 8 for additional metrics).

**Impact of budget $b$:** For the true GED, all possible mappings (factorial in the graph size) must be considered. Instead, GRAIL restricts this to $b$ mappings, where each mapping is generated by a program. In Fig. 3c, we plot how $b$ affects the upper bound. As shown, the upper bound converges at $\approx 15$ functions. Similar trends are observed in other datasets (see Fig. 5).

### 6.4. Interpretability: Case Study on IMDB

To shed light on the superior performance of GRAIL over GNN-based neural approximators, we analyze a graph pair from the IMDB dataset, where GRAIL shows the highest improvement over all baselines (Table 3). Figures 9 and 10 in the Appendix illustrate the graph structures, the edits made by GRAIL and its closest competitor GEDGNN, and the node similarity matrices generated by these algorithms. The program discovered by GRAIL-MIX, shown in Fig. 11, achieves the ground truth GED of 4, while GEDGNN predicts a GED of 20. This program assigns node similarity scores based on degree similarity and that of their neighbors. Since IMDB is unlabeled, feature similarity does not influence the results.

Examining GEDGNN's similarity matrix reveals a different score distribution compared to GRAIL. For instance, node 1 in Graph 1 has the second-highest similarity to nodes 2 and 9 in GRAIL-MIX, but GEDGNN assigns low similarity to these nodes, favoring nodes 0 and 6 instead. GRAIL-MIX's decision aligns with the similarity in their degrees (degree of 9 for node 1 in Graph 1 versus 7 for nodes 2 and 9 in Graph 2). In contrast, GEDGNN, as a neural network, operates as a black box. We hypothesize that the poor performance of GEDGNN and other GNN-based algorithms in IMDB is due to the dataset's unlabeled nature and high density, leading to oversquashing. (Giovanni et al., 2024).

We also qualitatively compare the GRAIL's heuristics with non-neural ones, and reason why GRAIL outperforms them in App. A.5.3.

## 7. Conclusions and Future Directions

This paper introduced a new paradigm of computing GED by leveraging LLMs to autonomously generate programs. Unlike traditional methods that rely on neural networks and require computationally expensive, NP-hard ground truth data, our method employs a self-evolutionary strategy to discover programs without any ground truth data. Remarkably, these programs not only surpass state-of-the-art methods on average but are also interpretable and demonstrate strong transferability across various datasets and domains. While our approach is demonstrated on GED computation, we believe it is generalizable to other combinatorial problems with similar constraints, both within and beyond graph-related tasks. An interesting direction for future work is to critically analyze the programs discovered by our method with domain experts and to develop mechanisms that facilitate closer cooperation between human and LLM agents.

## Acknowledgements

The authors extend their sincere gratitude to Alexander Novikov (Research scientist, Google DeepMind) for his thorough review of the initial draft of this work. His insightful comments and constructive feedback were instrumental in refining the manuscript and enhancing its clarity. Ananya Mathur thanks the IIT Delhi CSE Research Acceleration Fund for the generous travel grant given to her. Samidha Verma thanks Yardi School of Artificial Intelligence for supporting her research and Google for sponsoring her travel through the Google PhD Fellowship.

## Impact Statement

This work introduces a novel approach to Graph Edit Distance (GED) computation by leveraging large language models (LLMs) and evolutionary algorithms to generate interpretable programs for GED approximation. Unlike existing methods, our approach prioritizes transparency, interpretability, and cross-domain generalization while achieving state-of-the-art performance in approximation accuracy.

The societal implications of this work are significant. By addressing key limitations of neural approaches—such as their reliance on costly ground truth data, lack of interpretability, and domain-specific retraining—our method has the potential to make graph similarity computation more accessible and efficient across a variety of applications, including bioinformatics, social network analysis, and cheminformatics. Moreover, the transparency of program-based solutions could foster trust and reliability in critical domains where understanding the computation process is essential, such as healthcare or legal systems.

We do not foresee any ethical concerns arising out of our work.

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

# A. Appendix

## A.1. Neural GED Approximation Methods

- **GREED** (Ranjan et al., 2022): GREED utilizes a Siamese Graph Neural Network (GNN) to generate embeddings for graph pairs and learns to approximate the GED by preserving its metric properties. This is achieved by minimizing the RMSE between the norm of the embedding difference and the ground truth GED.

  We use the default hyperparameters used by the authors. Training uses a learning rate of 1e-3, weight decay of 1e-3, with a cyclic learning rate schedule (step_size_up/down = 2000) and early stopping patience of 5 cycles. The authors use an 8-layer Graph Isomorphism Network (GIN) (Xu et al., 2019). The hidden layer dimension is set to 64.

- **GEDGNN** (Piao et al., 2023): GEDGNN learns graph embeddings using a GNN followed by two computation heads, a cross matrix module that predicts the matching. Further, a combination of a cross matrix module and a neural tensor network, combined with the predicted matching matrix, is used to predict the GED value. GEDGNN does post-processing with a $k$-best matching algorithm for producing the edit path. Using this algorithm, they prune out bad matchings that have a larger edit path, enabling them to lower the bound for the GED estimate.

  We have used the parameter values as described by the authors. We use a 3-layered GIN with hidden dimensions 128, 64, and 32, respectively. 16 parameter matrices with size 32 x 32 are used in each cross matrix module. The default value of $k$ used during testing i.e. 100. Model training was done for 20 epochs, i.e., when the loss becomes smaller than 0.001. During training, the dropout rate is set to 0.5, the learning rate to 0.001, and the weight decay to 0.0005. GEDGNN employs a loss function that combines the matching loss and the GED value loss, with the latter weighted by a hyperparameter $\lambda$. As specified in the paper, $\lambda$ is set to 10 for the AIDS and Linux datasets and to 1 for the remaining datasets, which we adhere to in our implementation.

- **ERIC** (Zhuo & Tan, 2022): ERIC is a graph similarity model that learns node alignment implicitly through a regularizer, removing the need for a separate alignment module. It computes similarity using a Neural Tensor Network and MLP on graph-level embeddings, which are formed by concatenating outputs from a 5-layer GIN with 10-dimensional node embeddings. The model is trained using a combined loss that integrates both the regularizer and the similarity prediction. Training is performed using the Adam optimizer (learning rate 0.001, weight decay 0.0005), with a batch size of 256 and early stopping with a patience of 100 epochs.

- **H²MN** (Zhang et al., 2021): H²MN involves converting each input graph into a hypergraph to capture high-order structural relationships. It performs subgraph-level matching using hyperedges as subgraphs, coarsens the hypergraphs with a hyperedge pooling operator that is based on Personalized PageRank (Haveliwala, 2002). Then it employs a subgraph matching module, which compares important hyperedges from two graphs to capture fine-grained structural similarities. Finally embeddings are aggregated via a readout function and passed through an MLP to get the predicted GED value.

  We use the default hyperparameters used by the authors. All datasets use 3 hypergraph neural networks (HGNN) layers, 100-dimensional node embeddings, and a weight decay of 0.0005. Learning rate is set to 0.0001. Subgraph construction is based on a random walk of length 5 and a 2-hop neighborhood. The restart ratio for PageRank is 0.1, and the number of power iterations is 10. Each model is trained for 10,000 epochs with early stopping triggered if validation loss does not improve for 100 consecutive epochs.

- **GRAPHEDX** (Jain et al., 2024) : GRAPHEDX is composed of two main components: $\text{EMBED}_\theta$ and $\text{PERMNET}_\phi$. $\text{EMBED}_\theta$ includes a GNN module with 5 message-passing layers that generate 10-dimensional node embeddings using a GRU-based update function and a Linear-ReLU-Linear structure. After propagation, a separate Linear-ReLU-Linear network, computes 20-dimensional edge embeddings from the combined node features and adjacency indicators. $\text{PERMNET}_\phi$ then takes the final node embeddings from two graphs and processes them through a Linear-ReLU-Linear network to map them to a fixed size (10 for Linux, 20 for others). A similarity matrix is computed using $\mathcal{L}1$ distance, and 20 iterations of Sinkhorn normalization with temperature $\tau = 0.01$ are applied to generate a soft permutation matrix $\mathcal{P}$, which is used to compute the final soft edge alignment matrix $\mathcal{S}$.

## A.2. Proofs

### A.2.1. NP-HARDNESS OF EQ. 4

## Reduction to Prove NP-Hardness

We reduce the *Set Cover* problem to the given problem in polynomial time to demonstrate its NP-hardness.

Given a universe of elements $U = \{e_1, e_2, \ldots, e_n\}$, a collection of sets $\mathcal{S} = \{S_1, S_2, \ldots, S_m\}$ where $S_i \subseteq U$, and a budget $b$, the set cover problem seeks to determine if there exist $b$ sets $S_1, \ldots, S_b \in \mathcal{S}$ whose union covers all elements of $U$.

Given an instance of the set cover problem, we construct a bipartite graph $\mathcal{B} = (\mathcal{V}, \mathcal{U}, \mathcal{E}, \mathcal{W})$, where $\mathcal{V} = \mathcal{S}, \mathcal{U} = U$, and an edge $(S_i, e_j) \in \mathcal{E}$ exists if and only if $S_i$ covers $e_j$. Each edge $(S_i, e_j)$ has a weight:

$$w(S_i, e_j) = \begin{cases} 1 & \text{if } S_i \text{ covers } e_j, \\ 1 + \Delta & \text{if } S_i \text{ does not cover } e_j. \end{cases}$$

where $\Delta > 0$.

The objective is to select $b$ nodes from $\mathcal{V}$ (representing sets $\mathcal{S}$) such that Eq. 4 is minimized on graph $\mathcal{B}$.

If a *Set Cover* of size $b$ exists, then all $n$ elements can be covered by $b$ sets. This means if we select the corresponding nodes $\mathcal{A}^* \subseteq \mathcal{V}$, then every node in $\mathcal{U}$ will have at least one edge of weight 1 from some node in $\mathcal{A}^*$ incident on it. Hence, $\mathcal{J}(\mathcal{A}^*)$ will return a cumulative sum of $n$.

Conversely, if no *Set Cover* of size $b$ exists, then some elements will not be covered by the selected sets, and their corresponding nodes in $\mathcal{U}$ will have only edges of edge weights $1 + \Delta$ from nodes in $\mathcal{A}^*$.

Therefore, a solution to the *Set Cover* problem exists iff selecting the corresponding nodes $\mathcal{A}^* \subseteq \mathcal{V}$ leads to $\mathcal{J}(\mathcal{A}^*) = n$. Conversely, if $\mathcal{J}(\mathcal{A}^*) > n$, it implies that no $b$-set cover exists.

### A.2.2. MONOTONICITY AND SUBMODULARITY

**Lemma 1.** *Monotonicity:* $\mathcal{J}(\mathcal{A}) \leq \mathcal{J}(\mathcal{A}')$ *if* $\mathcal{A} \supseteq \mathcal{A}'$.

*Proof.* Since $\mathcal{J}(\mathcal{A}')$ computes minimum over all available mappings in $\mathcal{A}'$, the minimum can only reduce when additional mappings are added to form $\mathcal{A}$. □

**Lemma 2.** *Submodularity:* $\mathcal{J}(\mathcal{A}) - \mathcal{J}(\mathcal{A} \cup \{P\}) \leq \mathcal{J}(\mathcal{A}') - \mathcal{J}(\mathcal{A}' \cup \{P\})$.

*Proof.* We seek to show that the marginal reduction in $\mathcal{J}(\mathcal{A})$ when a program (mapping) $P$ is added to $\mathcal{A}$ is atmost as large as adding $P$ to its subset $\mathcal{A}'$. We establish this through *proof by contradiction*.

Let us assume

$$\exists \mathcal{A} \supseteq \mathcal{A}', \ \mathcal{J}(\mathcal{A}) - \mathcal{J}(\mathcal{A} \cup \{P\}) > \mathcal{J}(\mathcal{A}') - \mathcal{J}(\mathcal{A}' \cup \{P\}) \tag{7}$$

Due to the min operator in Eq. 5, Eq. 7 implies that the additional number of graph pairs where $P$ contributes to the minimum mapping is higher when added to $\mathcal{A}$ than when added to $\mathcal{A}'$. This creates a contradiction, since if $P$ contributes to the minimum of a graph pair in $\mathcal{A} \cup \{P\}$, then it guaranteed to contribute to the minimum for the same pair in $\mathcal{A}' \cup \{P\}$ as well. □

## A.3. Experiments

### A.3.1. SETUP

All experiments ran on a machine equipped with an Intel Xeon Gold 6142 CPU @1GHz and a GeForce GTX 1080 Ti GPU. While non-neural methods and GRAIL run on the CPU, neural baselines exploit the GPU. For the LLM, we use Gemini 1.5 Pro. In particular, we have used the initial stable version of Gemini 1.5 Pro, i.e., gemini-1.5-pro-001, which was released on May 24, 2024.

**Hyper-parameters:** Table H lists the hyper-parameters used for GRAIL. $k$ stands for the number of functions per response

```python
def priority_v0(graph1: list[list[int]], graph2: list[list[int]], weights: list[list[float]]) -> list[list[float]]:
    """
    Computes the Graph Edit Distance (GED), a measure of the dissimilarity between two graphs.
    GED is defined as the minimum number of operations required to transform one graph into another.
    The primary operations considered in GED calculations include:
    - **Node Insertion/Deletion:** Adding or removing a node incurs a cost of +1.
    - **Edge Insertion/Deletion:** Adding or removing an edge between two nodes incurs a cost of +1.
    - **Node Relabeling:** Modifying the label of a node (if labels are present) adds a cost of +1 for each mismatch.

    Args:
        graph1: The adjacency matrix of the first graph.
        graph2: The adjacency matrix of the second graph.
        weights: A weight matrix representing the initial probabilities of mapping nodes between
                 `graph1` and `graph2`. Each entry is a probability value, where a higher value
                 indicates a higher likelihood and similarity of mapping nodes. The size of the
                 weight matrix is determined by the maximum number of nodes in both graphs squared.
    Returns:
        A refined weight matrix (list[list[float]]) where each entry represents the probability of
        a node in `graph1` being mapped to a node in `graph2` in a way that minimizes the overall
        graph edit distance.
    """
    max_node = len(graph1)
    weights = [[0.0] * max_node for _ in range(max_node)]
    return weights

def priority_v1(graph1: list[list[int]], graph2: list[list[int]], weights: list[list[float]]) -> list[list[float]]:
    """ Improved version 'priority_v0' """
    n1 = len(graph1)
    n2 = len(graph2)
    max_node = max(n1, n2)
    refined_weights = [[0.0] * max_node for _ in range(max_node)]

    for i in range(n1):
        for j in range(n2):
            neighbor_similarity = 0
            for k in range(n1):
                for l in range(n2):
                    if (graph1[i][k] == 1) and (graph2[j][l] == 1):
                        neighbor_similarity += weights[k][l]
            refined_weights[i][j] = neighbor_similarity

    return refined_weights

def priority_v2(graph1: list[list[int]], graph2: list[list[int]], weights: list[list[float]]) -> list[list[float]]:
    """ Improved version 'priority_v1' """
```

Problem description

Task specification

Top-*k* programs(*k*=2)

Function header for *k*+1ᵗʰ program

*Figure 4.* Example of an input prompt to GRAIL

generated by the LLM and $b$ is the function budget employed for submodularity while training. We decided to use $k$ as 2, since with greater values of k, we observed no significant improvement in quality metrics. This was also observed in FunSearch (Romera-Paredes et al., 2024). For the function budget $b$, we observed that a value of 15 was good enough for most datasets (Refer Fig. 5).

### A.4. Datasets

The semantics of these datasets are as follows:

• **ogbg-molhiv and ogbg-molpcba:** These are chemical compound datasets, with each graph representing a molecule.

| Hyper-parameter | Value |
|---|---|
| $k$ | 2 |
| $b$ | 15 |
| number of islands | 5 |
| temperature | 0.99 |
| Algorithm for bipartite matching | Neighbor-biased mapper (He & Singh, 2006) |

*Table H.* Hyper-parameters used for GRAIL

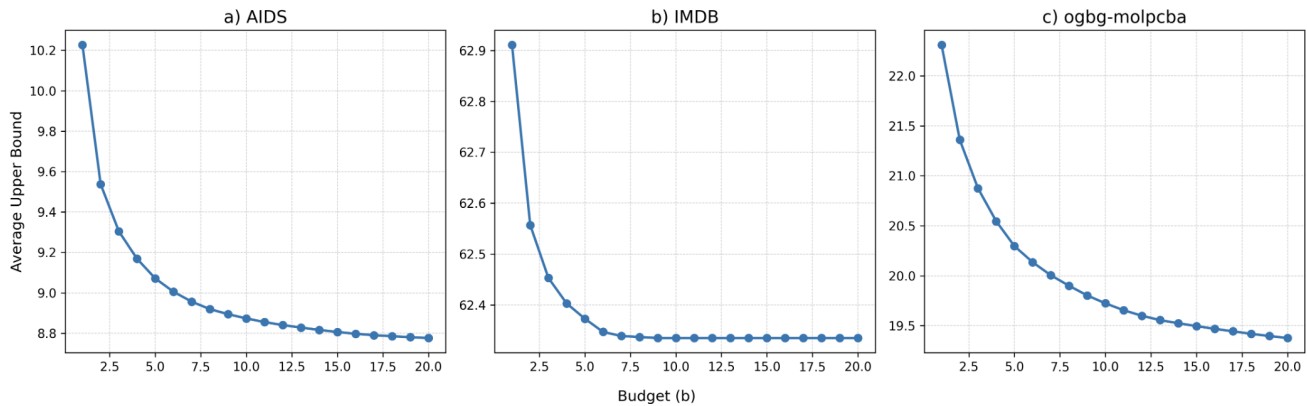

*Figure 5.* Avg. Upper Bound vs function budget (b) for submodular greedy selection

Nodes in these graphs correspond to atoms and are labeled with their atomic numbers, while edges denote the chemical bonds between atoms. These datasets vary in size and complexity, with a rich diversity of molecular structures, enabling us to test the robustness and generalizability of our method.

- **ogbg-code2:** This dataset comprises a vast collection of Abstract Syntax Trees (ASTs) generated from nearly 450,000 Python method definitions. Each graph in this dataset represents an AST, with nodes labeled from a predefined set of 97 categories, capturing various syntactic constructs within the methods. These graphs are considered undirected, simplifying the representation while preserving structural relationships.

- **ogbg-ppa:** This dataset includes undirected protein association neighborhoods extracted from protein-protein interaction networks of 1,581 species(Szklarczyk et al., 2019) across 37 diverse taxonomic groups. To build these neighborhoods, 100 proteins were randomly selected from each species, and 2-hop protein association neighborhoods were constructed around each selected protein(Zitnik et al., 2019). In these graphs, proteins are represented as nodes, and edges indicate biologically relevant associations between them.

- **AIDS:** This dataset is a collection of graphs sourced from the AIDS antiviral screen database, each graph representing a chemical compound's molecular structure. These graphs are labeled, capturing meaningful properties of the compounds, and are compact in size, containing no more than 10 nodes.

- **Linux:** A collection of program dependence graphs where nodes correspond to statements and edges indicate dependencies between statements. The graph sizes in this dataset are also limited to 10 nodes. This dataset is unlabeled and was introduced in (Wang et al., 2012).

- **IMDB:** This dataset consists of ego-networks of actors and actresses who have shared screen time in movies. Each graph represents an ego-network where the nodes correspond to individuals (actors or actresses), and the edges denote shared appearances in films. This dataset is unlabeled and was introduced in (Yanardag & Vishwanathan, 2015).

### A.4.1. GROUND-TRUTH DATA GENERATION

We employ MIP-F2 (Lerouge et al., 2017b) to generate ground truth GED. MIP-F2 returns the lower and upper bounds of GED. We compute these bounds with a time limit of 600 seconds per pair. Pairs with equal lower and upper bounds are included in the ground truth.

### A.4.2. METRICS

We use the following two metrics to quantify accuracy:

- **RMSE:** Evaluates the prediction accuracy by measuring the disparities between actual and predicted values. For $n$ graph pairs, it is defined as:

$$\sqrt{\frac{1}{n} \sum_{i=1}^{n} (\text{true-ged}_i - \text{pred-ged}_i)^2}$$

- **Exact Match Ratio:** Represents the proportion of graph pairs where the predicted GED exactly matches the actual GED. For $n$ graph pairs, it is defined as:

$$\frac{1}{n} \sum_{i=1}^{n} \mathbb{I}\big(\text{true-ged}_i = \text{pred-ged}_i\big)$$

where $\mathbb{I}(\cdot)$ is an indicator function that returns 1 if the condition inside is true, and 0 otherwise. A higher Exact Match Ratio indicates better predictive accuracy at the individual graph pair level.

| Method | AIDS | Linux | IMDB | ogbg-molhiv | ogbg-code2 | ogbg-molpcba |
|--------|------|-------|------|-------------|------------|--------------|
| GRAIL | 0.83 | $\approx 1$ | 0.99 | 0.18 | 0.11 | 0.12 |

*Table 9.* EMR results of GRAIL for all datasets. Values in the range (0.99,1) are denoted as $\approx 1$

### A.5. GRAIL vs. Non-neural Methods

#### A.5.1. COMPARISON OF GRAIL WITH ENSEMBLE OF NON-NEURAL METHODS

Similar to GRAIL, several non-neural heuristics also consider multiple solutions. Specifically, branch-and-bound methods (e.g., BRANCH) and MIP approaches (LP-GED-F2, ADJ-IP, COMPACT-MIP) explore multiple candidate solutions that satisfy the given constraints, selecting the one with the tightest upper bound.

In this experiment, we compare GRAIL with an ensemble of non-neural GED approximators. Specifically, we select the 15 non-neural heuristics from (Blumenthal et al., 2020), based on their tightness of upper bounds. The set of 15 heuristics used in the ensemble includes, ADJ-IP, NODE, LP-GED-F2, BRANCH-TIGHT, COMPACT-MIP, IPFP, BRANCH-CONST, BRANCH-COMPACT, BRANCH-FAST, BP-BEAM, BRANCH, K-REFINE, STAR, REFINE, and F1

For each graph pair, we apply all 15 non-neural heuristics and choose the minimum computed distance. As shown in Table 10, GRAIL continues to outperform this ensemble approach. It should also be noted that GRAIL is approximately **168x** faster than the ensemble.

| Method | AIDS | LINUX | IMDB | ogbg-molhiv | ogbg-code2 | ogbg-molpcba |
|--------|------|-------|------|-------------|------------|--------------|
| Non-Neural Ensemble | 0.73 | 0.23 | 0.72 | 3.13 | 7.10 | 3.66 |
| GRAIL | 0.57 | 0.13 | 0.55 | 2.96 | 4.22 | 3.18 |

*Table 10.* RMSE of non-neural ensemble and GRAIL across datasets. The best RMSE has been highlighted with green color

#### A.5.2. COMPARISON OF INDIVIDUAL FUNCTIONS' PERFORMANCE

In Table 11, we report the best, worst, and average RMSE for the 15 heuristics identified by GRAIL, denoted as Min, Max, and Avg, respectively. Similarly, we provide the Min, Max, and Avg RMSEs for the ensemble of non-neural heuristics as specified in section A.5.1.

Notably, the best non-neural heuristic outperforms the best function identified by GRAIL in 4 out of 6 datasets. However, this result aligns with our expectations, as GRAIL does not aim to find a single optimal function with the lowest RMSE. Instead, leveraging a submodular loss function, it seeks a diverse set of complementary heuristics that collectively minimize RMSE across the training set of graph pairs.

#### A.5.3. QUALITATIVE COMPARISON OF DISCOVERED HEURISTICS VS. EXISTING NON-NEURAL HEURISTICS

The heuristics discovered by GRAIL are fundamentally different from the top-performing heuristics in (Blumenthal et al., 2020). Specifically:

- NODE and BRANCH transform GED into a linear sum assignment problem.

| Method / Datasets | AIDS | Linux | IMDB | ogbg-molhiv | ogbg-code2 | ogbg-molpcba |
|---|---|---|---|---|---|---|
| Min GRAIL | 1.90 | 0.49 | 3.19 | 6.23 | 6.48 | 6.61 |
| Max GRAIL | 3.33 | 1.32 | 7.48 | 8.27 | 17.97 | 9.40 |
| Avg. GRAIL | 2.57 | 0.96 | 5.04 | 7.16 | 12.10 | 7.96 |
| Min Non-Neural | 0.85 | 0.23 | 3.90 | 4.50 | 8.34 | 4.94 |
| Max Non-Neural | 15.23 | 6.72 | 69.45 | 24.97 | 92.67 | 29.22 |
| Avg Non-Neural | 7.62 | 4.08 | 16.99 | 16.52 | 39.24 | 19.37 |

*Table 11.* RMSE of individual functions discovered by GRAIL compared to non-neural methods across datasets. The best Min. RMSE has been highlighted with green color

- LP-GED-F2, ADJ-IP, and COMPACT-MIP use mixed integer programming.

- IPFP reduces GED to the quadratic assignment problem (QAP), which is also NP-hard and relies on heuristics to approximate the QAP.

In contrast, GRAIL's discovered heuristics compute similarity between all node pairs across the two graphs followed bipartite matching. This similarity is determined by measuring the distance between various vertex invariants—such as node labels, degree, etc.—within their $l$-hop neighborhoods. The heuristics differ in their choice of vertex invariants, distance functions, and $l$.

The heuristics discovered by GRAIL can, in principle, replace those used by NODE or BRANCH. Qualitatively, NODE does not consider edge information, and the cost matrix is primarily constructed based on the cost of node relabelling, insertion, or deletion. BRANCH, on the other hand, takes the node labels and the labels of the edge end points into consideration when constructing the cost matrix.

In contrast, the heuristics discovered by GRAIL are more general. They consider the label information of nodes and edge end points as well as compute the cost based on vertex invariants such as the degree distributions of the nodes to be matched and their k-hop neighborhoods. Finally, normalization and weighing factors are also introduced while constructing the cost matrix, as seen in Fig. 11. Thus, the cost matrix is perhaps more rich in terms of assessing the cost of aligning two nodes from a graph pair.

The complete repository of discovered heuristics is available at https://github.com/idea-iitd/Grail inside **src/discovered_programs**.

### A.6. Efficiency Analysis

The training and inference time analysis is shown in Fig. 6.

**Training Time:** From Fig. 6a, we observe that GRAIL is significantly more efficient than the neural baselines. Neural methods require NP-hard ground truth training data, which involves extensive computation times of up to 15 days. Additionally, note that GRAIL-MIX requires training only once while performing on par with GRAIL and neural baselines in terms of approximation error (see Table 3).

**Inference Time:** We compare the inference time of GRAIL with the top three neural and non-neural methods from Table 3, as shown in Fig. 6b. At the onset, we point out that while GRAIL infers on CPUs and provides the node mapping in addition to the predicted GED, neural baselines rely on GPUs and only provide the GED. Hence, neural methods have a lower computational workload while having access to more powerful computational resources. Results indicate that GRAIL achieves faster inference times than neural baselines for smaller and sparser datasets, such as AIDS and Linux. However, inference times increase for larger and denser datasets, such as IMDB and ogbg, due to the computational overhead of computing mappings. The maximum recorded inference time is 94.6 seconds for 968 graph pairs in the ogbg-code2 dataset ($\sim 0.1$ seconds per pair), which remains reasonable considering the various advantages of GRAIL, including its independence from ground truth data, one-time training for GRAIL-MIX, and strong generalization capabilities. Furthermore, the efficiency of GRAIL's programs can be further improved through human intervention or translation to more efficient languages, such as C.

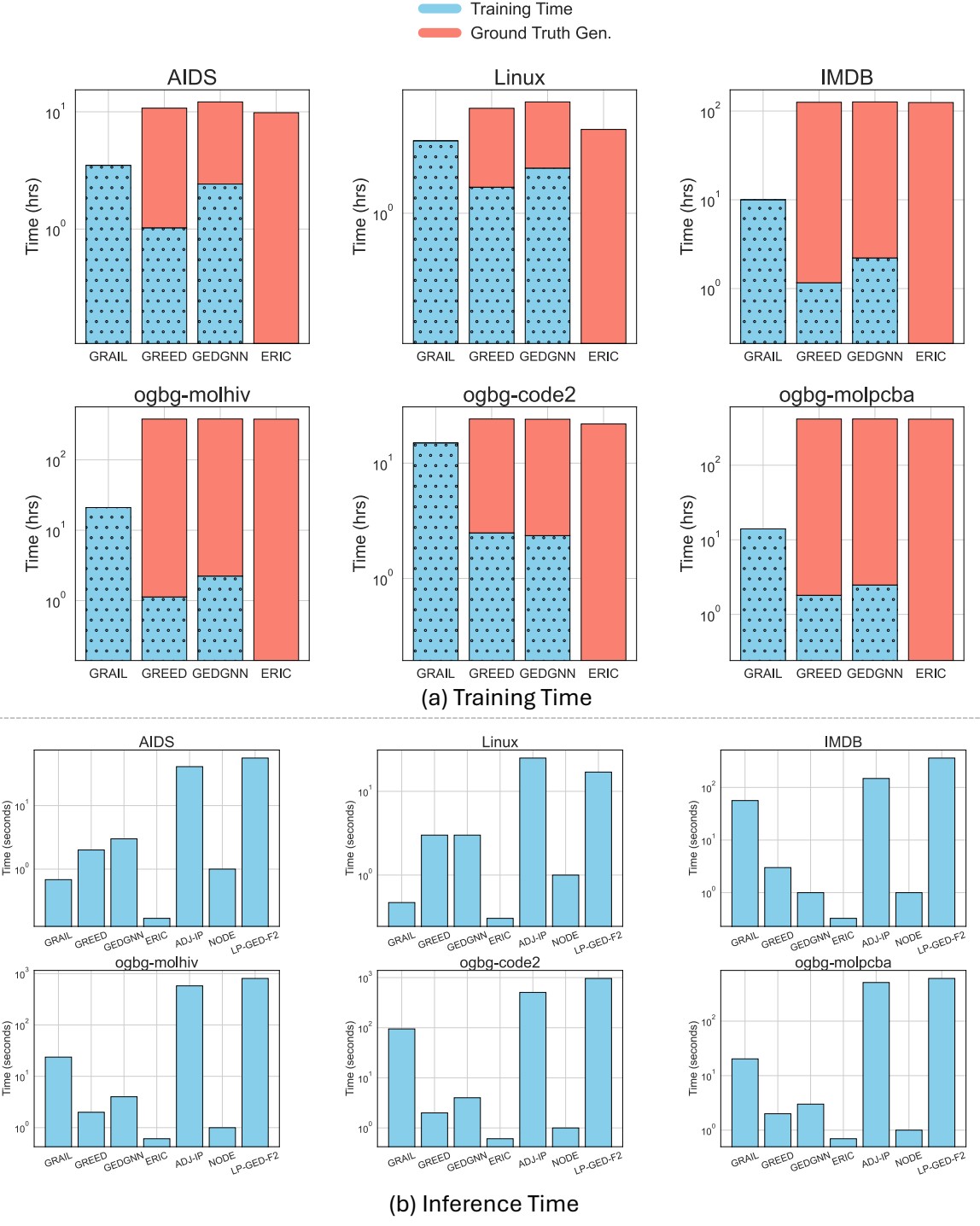

Figure 6. (a) Training Time: Comparison of GRAIL with the top-3 neural methods in Table 3. (b) Inference Time: Comparison of GRAIL with the top-3 neural and non-neural methods in Table 3. The top-3 methods have been selected based on avg. ranks.

**Note:** Ground truth generation time is the same for a dataset (9 hrs 43 min : AIDS, 3 hrs 25 min : Linux, 124 hrs 25 min : IMDB, 379 hrs 19 min : ogbg-molhiv, 21 hrs 42 min : ogbg-code2, 414 hrs 30 min : ogbg-molpcba) for all neural methods, but appears to be different in the plots due to log scale conversion.

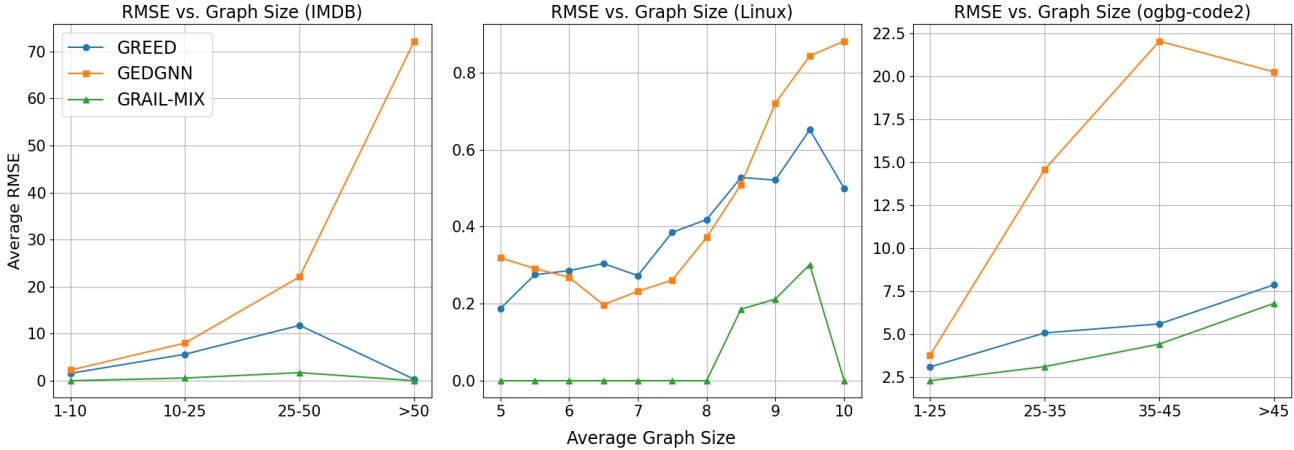

*Figure 7.* Avg. RMSE vs. Avg. Graph Size comparison on IMDB, Linux and ogbg-code2 datasets. GRAIL-MIX outperforms the best baselines at both smaller and larger graph sizes. The rate of increase of error is lower for GRAIL-MIX as opposed to GREED and GEDGNN with increasing average graph size.

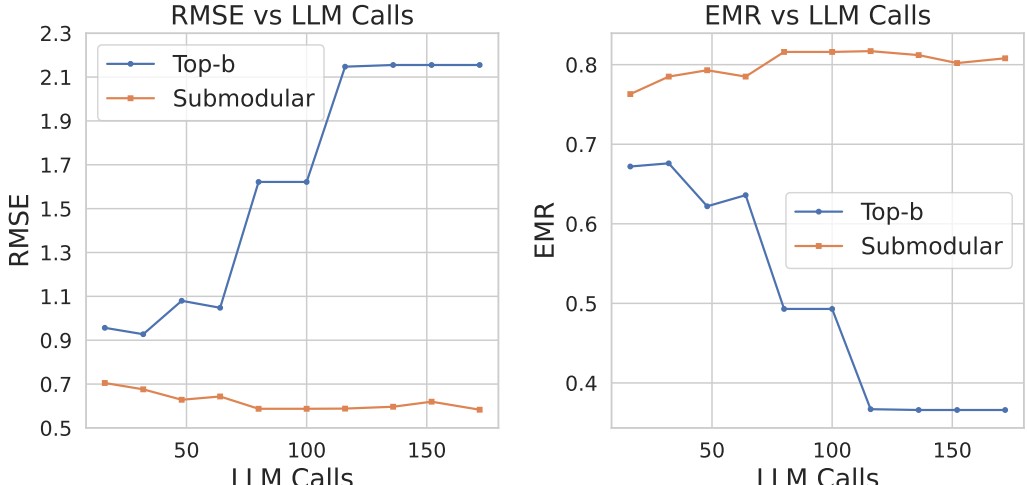

*Figure 8.* Performance comparison of Top-*b* vs. Greedy Submodular on the test set of AIDS dataset with an increasing number of LLM calls.

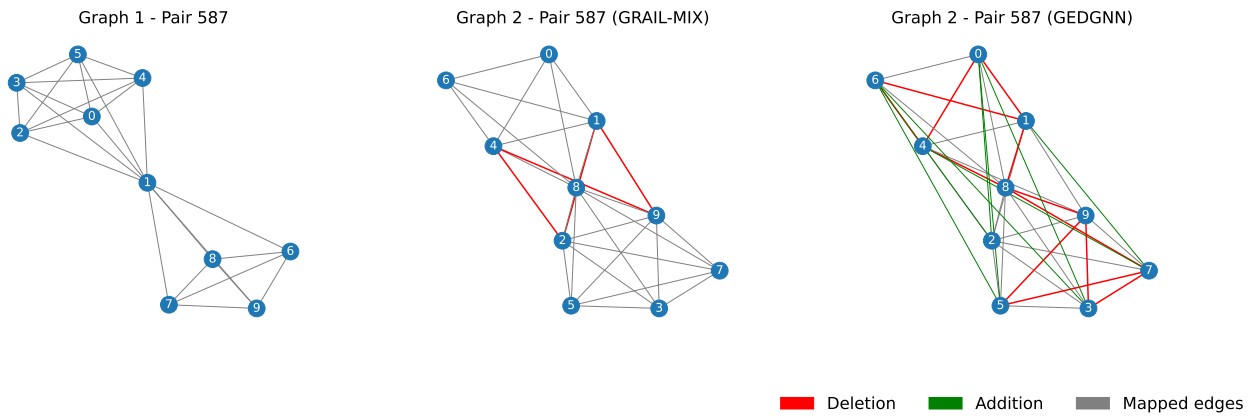

*Figure 9.* IMDB Case Study: The left-most graph represents Graph 1, while the middle and right-most graphs depict Graph 2 with predicted edits from GRAIL-MIX (Fig: 11) and GEDGNN, respectively. The red and green edges in each graph indicate the edge edits predicted by both methods. Ground Truth GED:4, GRAIL-MIX GED:4, GEDGNN Mapping's GED: 20.

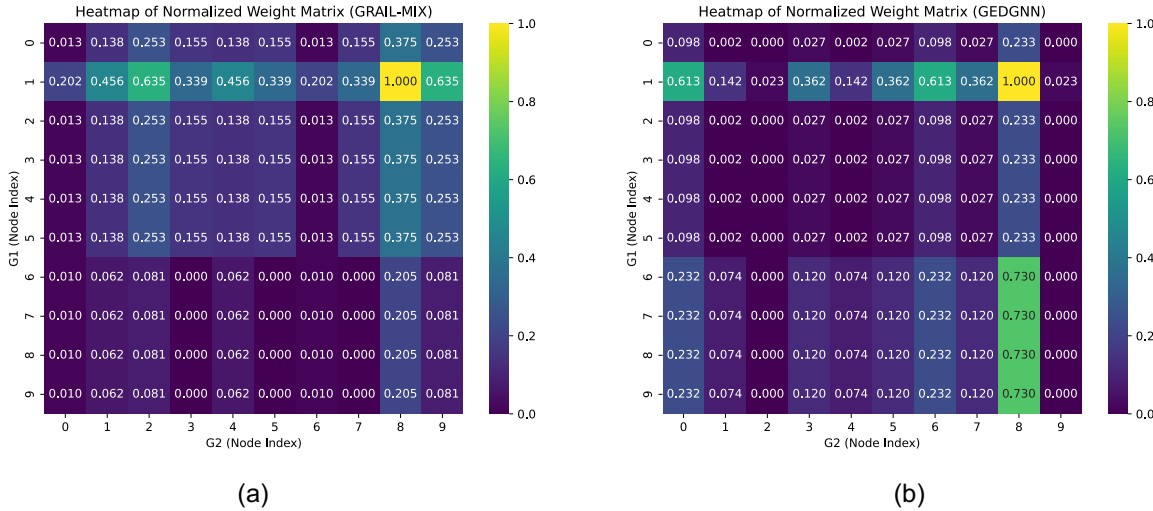

*Figure 10.* IMDB Case Study: Heatmap of weight matrix generated by (a) GRAIL-MIX (Fig: 11) and (b) GEDGNN

```python
def priority_v1(graph1: list[list[int]], graph2: list[list[int]], weights: list[list[int]]) -> list[list[float]]:
    n1 = len(graph1)
    n2 = len(graph2)
    max_node = max(n1, n2)

    graph1_np = np.array(graph1)
    graph2_np = np.array(graph2)
    weights_np = np.array(weights)
    degrees1 = graph1_np.sum(axis=1)
    degrees2 = graph2_np.sum(axis=1)
    refined_weights = np.zeros((max_node, max_node), dtype=float)

    for i in range(n1):
        neighbors_i = np.where(graph1_np[i] == 1)[0]
        for j in range(n2):
            score = 0.0
            if i < weights_np.shape[0] and j < weights_np.shape[1]:
                score += weights_np[i, j]

            degree_diff = abs(degrees1[i] - degrees2[j])
            score += 1 / (1 + degree_diff) # high score if degree diff is low
            neighbors_j = np.where(graph2_np[j] == 1)[0] #neighbours of j
            common_neighbors = 0
            if neighbors_i.size > 0 and neighbors_j.size > 0:
                neighbor_weights = weights_np[neighbors_i.reshape(-1, 1), neighbors_j]
                common_neighbors = neighbor_weights.sum()
            score += common_neighbors

            neighbor_similarity = 0
            if neighbors_i.size > 0 and neighbors_j.size > 0:
                degree_diffs = np.abs(degrees1[neighbors_i.reshape(-1, 1)] - degrees2[neighbors_j])
                neighbor_similarity = (neighbor_weights * (1 / (1 + degree_diffs))).sum()
            score += neighbor_similarity
            refined_weights[i, j] = score

    total_score = refined_weights[:n1, :n2].sum()

    if total_score > 0:
        refined_weights[:n1, :n2] /= total_score
    else:
        refined_weights[:n1, :n2] = 1 / (n1 * n2) if (n1 * n2) > 0 else 0

    return refined_weights.tolist()
```

**Initializing score with feature-based weight similarity of nodes *i* (*graph 1*) and *j* (*graph 2*).**

**Degree comparison of nodes *i* (*graph 1*) and *j* (*graph 2*).**

**Initial Feature-based similarity of neighbors of nodes *i* (*graph 1*) and *j* (*graph 2*.**

**Weighted degree similarity of neighbors of nodes *i* (*graph 1*) and *j* (*graph 2*.**

**Aggregation and normalization**

*Figure 11.* IMDB Case Study: Program discovered by GRAIL-MIX that has minimum individual RMSE on IMDB dataset.

