# OpenReview forum: "GRAIL: Graph Edit Distance and Node Alignment using LLM-Generated Code"
_ICML.cc/2025/Conference — ICML 2025 poster_

### Official Review · Reviewer_vS7D · 2025-03-14

**Overall Recommendation:** 3

**Summary:**

This paper presents an evolutionary-based algorithm that generates better programs by large language models to calculate heuristics scores for graph edit distance. With generated scores, the LLM-discovered algorithm achieved better heuristic prediction than other GED methods.

**Claims And Evidence:**

(+) All claims seem well-grounded by empirical evaluations on various graph datasets.

**Essential References Not Discussed:**

No

**Experimental Designs Or Analyses:**

(+) Most numerical evaluations are sound and the improvement is solid and impressive.

(-) One concern is that the LLM-generated algorithm allows 15 different scoring functions while most non-neural GED heuristics contain only one scoring function, the following two baselines should be considered to decouple the improvement from simply adding more scoring functions:
* Take the top-15 performing GED heuristics from [Blumenthal et al. 2020] and evaluate the performance of that "ensembled" algorithm
* Take each scoring function generated by LLM and compare that with traditional heuristics.

(-) Also, it is worth doing a qualitative evaluation of how the generated heuristics look like and how different are they from those published ones. It is very likely that those GED heuristics papers are part of the LLM training dataset, therefore showing quantitatively that LLM discovered something different from existing literature is important for the novelty of this manuscript.

**Methods And Evaluation Criteria:**

(+) Most peer methods and evaluation criteria make sense for the problem.

**Other Comments Or Suggestions:**

N/A

**Other Strengths And Weaknesses:**

N/A

**Questions For Authors:**

I would be more than glad to see the authors' responses to my questions on 1 scoring function vs 15 and potential qualitative evaluations. That is the critical limitation for me to be more positive about this submission or turn negative.

**Relation To Broader Scientific Literature:**

(+) This paper has its novelty due to the fundamental nature of GED (and other NP-hard problems) and tackling them by LLM-generated algorithms is interesting.

**Theoretical Claims:**

(+) I checked the theoretical claims and they are straightforward and align with my intuition. Therefore I did not check the detailed proofs.

---

> ### Author Rebuttal · Authors · 2025-03-29
>
> **W1(a) Take the top-15 performing GED heuristics from \[Blumenthal et al. 2020\] and evaluate the performance of that "ensembled" algorithm**
>
> **Ans:** First, we would like to clarify that several non-neural heuristics also consider multiple solutions. Specifically, branch-and-bound methods (e.g., **BRANCH**) and MIP approaches (**LP-GED-F2, ADJ-IP, COMPACT-MIP**) explore multiple candidate solutions that satisfy the given constraints, selecting the one with the tightest upper bound.
>
> Nonetheless, we have conducted the suggested experiment. Specifically, we select the 15 non-neural heuristics from Blumenthal et al., based on their tightness of upper bounds. For each graph pair, we applied all 15 heuristics and chose the minimum computed distance. As shown in the results below, **GRAIL continues to outperform this ensemble approach**. We also note that Grail is more than $\approx 168$-times faster.
>
> 1. ADJ-IP
> 2. NODE
> 3. LP-GED-F2
> 4. BRANCH-TIGHT
> 5. COMPACT-MIP
> 6. IPFP
> 7. BRANCH-CONST
> 8. BRANCH-COMPACT
> 9. BRANCH-FAST
> 10. BP-BEAM
> 11. BRANCH
> 12. K-REFINE
> 13. STAR
> 14. REFINE
> 15. F1
>
> ### RMSE
> |Dataset|Ensemble Non-neural|GRAIL|
> |-|-|-|
> | AIDS|0.73|**0.57**|
> | LINUX|0.23|**0.13**|
> | IMDB|0.72|**0.55**|
> | molhiv|3.13|**2.96**|
> | code2|7.10|**4.22**|
> | molpcba|3.66|**3.18**|
>
> **W1(b) -   Take each scoring function generated by LLM and compare that with traditional heuristics.**
>
> **Ans:** "In the table below, we report the best, worst, and average RMSEs for the 15 heuristics identified by Grail, denoted as Min, Max, and Avg, respectively. Similarly, we provide the Min, Max, and Avg RMSEs for the ensemble of non-neural heuristics.
>
> Notably, the best non-neural heuristic outperforms the best function identified by Grail in 4 out of 6 datasets. However, this result aligns with our expectations, as Grail does not aim to find a single optimal function with the lowest RMSE. Instead, leveraging a submodular loss function, it seeks a diverse set of complementary heuristics that collectively minimize RMSE across the training set of graph pairs."
>
> |Dataset|AIDS|Linux|IMDB|molhiv|code2|molpcba|
> |-|-|-|-|-|-|-|
> | **Min** Grail | 1.90 | 0.49| 3.19| 6.23| 6.48| 6.61|
> | **Max** Grail|3.33 |1.32| 7.48|8.27|17.97|9.40|
> | **Avg.** Grail| 2.57|0.96| 5.04|7.16|12.10|7.96|
> | **Min** Non-Neural |0.85|0.23| 3.90 |4.50|8.34 |4.94|
> | **Max** Non-Neural|15.23|6.72|69.45|24.97|92.67|29.22|
> | **Avg** Non-Neural|7.62|4.08|16.99|16.52|39.24 |19.37|
>
> **W2(a):** **Qualitative evaluation of how the generated heuristics look like and how different are they from those published ones.**
>
> **Response:**  The heuristics discovered by Grail are fundamentally different from the top-performing heuristics in Blumenthal et al. Specifically:
>
> - **NODE** and **BRANCH** transform GED into a linear sum assignment problem.
> - **LP-GED-F2, ADJ-IP,** and **COMPACT-MIP** use mixed integer programming.
> - **IPFP** reduces GED to the quadratic assignment problem (QAP), which is also NP-hard and relies on heuristics to approximate the QAP.
>
> In contrast, our discovered heuristics compute similarity between all node pairs across the two graphs followed bipartite matching. This similarity is determined by measuring the distance between various vertex invariants—such as node labels, degree, etc.—within their $l$-hop neighborhoods. The heuristics differ in their choice of vertex invariants, distance functions, and $l$.
>
> We provide one example in **Fig. 11**. The complete repository of discovered heuristics is available at https://anonymous.4open.science/r/GRAIL-FEE5/ inside src/discovered_programs.
>
> **W2(b). It is very likely that those GED heuristics papers are part of the LLM training dataset, therefore showing quantitatively that LLM discovered something different from existing literature is important for the novelty of this manuscript.**
>
> **Response:** While the possibility of the heuristics being part of the training data can't be ruled out, we hypothesize that if the LLM has seen them during training, then it should be able to directly retrieve the programs without reducing GED to a bipartite matching problem, and then prompt tuning.
>
> To test this hypothesis, we prompted the LLM to generate code for GED directly, rather than predicting the weights of the bipartite graph. As shown below (**Direct**), the results deteriorate significantly, suggesting that the LLM alone is unable to retrieve effective heuristics without our structured formulation.
>
> ||DATASET|AIDS|Linux|IMDB|molhiv|code2|molpcba|
> |-|-|-|-|-|-|-|-|
> | **RMSE**|**GRAIL**|**0.57**|**0.13**|**0.55**|**2.96**|**4.22**|**3.18**|
> | |  **Direct fns (15)**|7.95|2.81|112.64| 7.47| 25.68| 9.71|
> | **EMR**|**GRAIL**|0.8 |~1|~1| 0.2| 0.12|0.12|
> | | **Direct fns (15)**| 0.01| 0.14| 0.09|0.02|0|0.02|
> ---------
> **Summary:** We appreciate the reviewer’s insightful suggestions and have incorporated the recommended experiments to further validate our approach. We hope these results clarify the merits of our work.

---

> > ### Comment · Reviewer_vS7D · 2025-04-02
> >
> > Thanks for the great work, I have no more questions; I would be in favor of accepting this manuscript.
> >
> > I would like to raise one minor point that solving a "bipartite matching" should be the same as solving a "linear sum assignment", therefore the LLM-generated heuristics could be considered as in-place replacements of the similarity matrix in linear assignment problems. A qualitative study showing that LLM _discovered_ better scoring functions than NODE or BRANCH heuristics would be very interesting to the community.

---

> > > ### Author Response · Authors · 2025-04-06
> > >
> > > Thank you for your encouraging feedback and for motivating us to further improve our work.
> > >
> > > The observation is indeed correct that LLM-generated heuristics could, in principle, replace those used by NODE or BRANCH. Qualitatively, NODE does not consider edge information, and the cost matrix is primarily constructed based on cost of node relabelling, insertion or deletion. BRANCH, on the other hand, takes the node labels and the labels of the edge end points into consideration when constructing the cost matrix. In contrast, the heuristics discovered by GRAIL are more general. They consider the label information of nodes, and edge end points as well as compute the cost based on vertex invariants such as the degree distributions of the nodes to be matched and their k-hop neighborhoods. Finally, normalization and weighing factors are also introduced while constructing the cost matrix as seen in Figure 11. Thus, the cost matrix is perhaps more rich in terms of assessing the cost of aligning two nodes from a graph pair.
> > >
> > > Lastly, if you feel that the revisions have strengthened our manuscript, we would be grateful for a reassessment of the rating.

---

### Official Review · Reviewer_cLnD · 2025-03-15

**Overall Recommendation:** 3

**Summary:**

In this paper, the authors introduced a new paradigm of computing GED by leveraging LLMs to autonomously generate programs. Unlike traditional methods that rely on neural networks and require computationally expensive, NP-hard ground truth data, proposed method employs a self-evolutionary strategy to discover programs without any ground truth data.

**Claims And Evidence:**

Yes, the claims made in the submission are supported by clear and convincing evidence.

**Essential References Not Discussed:**

No

**Experimental Designs Or Analyses:**

Yes, the experimental designs and analysis of the paper are reasonable.

**Methods And Evaluation Criteria:**

I believe that the method in this paper is relatively reasonable.

**Other Comments Or Suggestions:**

No

**Other Strengths And Weaknesses:**

Strengths
1.  The structure of the paper is clear and easy to follow.
2.  The method is effective in comparison to baseline methods.

Weaknesses
1. Is the role of the LLM merely to generate code programs? What are the advantages of this approach compared to manually collecting a series of graph edit distance programs?
2. In Table 3 and Table 4, GRAIL does not perform optimally on ogbg-molpcba. It is recommended that the authors provide reasons and insights for this observation.

**Questions For Authors:**

Please refer to the weaknesses section.

**Relation To Broader Scientific Literature:**

No

**Theoretical Claims:**

The paper does not include theoretical proofs, so there is no need to verify the correctness of the theoretical claims in the paper.

---

> ### Author Rebuttal · Authors · 2025-03-29
>
> **W1: Is the role of the LLM merely to generate code programs? What are the advantages of this approach compared to manually collecting a series of graph edit distance programs?**
>
> **Ans:** Yes, the LLM's role is to generate code as  responses to strategically curated prompts. This approach offers several advantages over manually collecting heuristics.
>
> * First, the LLM can explore an essentially *infinite space of heuristics*, far beyond what is feasible by human design alone.
> * Second, it enables *rapid iterative improvements* through prompt-tuning and feedback, which greatly accelerates the development cycle compared to the slow, trial-and-error process of manual design.
> * Finally, there is no comprehensive repository of human-devised heuristics for GED—the most extensive benchmark we know is by Blumenthal et al.—and our LLM-generated methods not only broaden the search space but also consistently outperform these limited manual methods.
>
>
> **W2: In Table 3 and Table 4, GRAIL does not perform optimally on ogbg-molpcba. It is recommended that the authors provide reasons and insights for this observation.**
>
> **Ans:** In this dataset, GRAIL is outperformed by GREED. GRAIL constructs a bipartite graph between node pairs across graphs, where edge weights encode similarity based on topological and node label similarities within their $l$-hop neighborhoods. The LLM-generated code computes these similarities, with $l=2$ fixed across all datasets.
>
> In contrast, GREED employs _jumping knowledge_, concatenating node embeddings from multiple hops (1 to 7), allowing an MLP to learn which depth is most effective for GED approximation. If we adopt a similar strategy in GRAIL—computing GED using $l=1,2,3$ and selecting the minimum (as the tightest upper bound corresponds to the minimum)—GRAIL achieves performance competitive with GREED. The detailed results are presented below.
>
> This finding suggests that, in ogbg-molpcba, using a uniform topological depth for all nodes is suboptimal for GED approximation. Adapting the neighborhood depth dynamically can significantly improve accuracy.
>
> ### RMSE values of Grail at various topological depths compared to the RMSE of GREED, which is 2.48
> Depth | GRAIL |
> ---|---|
> $l=1$ | 3.18
> $l=1,2$ | 2.73
> $l=1,2,3$ | 2.68
>
> ---------------
>
> **Summary:** We appreciate the reviewer’s constructive feedback and the opportunity to improve our work. We hope that our additional clarifications effectively address the outstanding concerns. If the reviewer finds our manuscript strengthened by these revisions, we would be grateful for a reassessment of our work’s rating.

---

### Official Review · Reviewer_isZf · 2025-03-19

**Overall Recommendation:** 3

**Summary:**

This paper presents a novel paradigm for computing Graph Edit Distance (GED) by harnessing the capabilities of LLMs to autonomously generate executable programs. Departing from conventional approaches that depend on neural networks and computationally intensive, NP-hard ground truth data, our methodology adopts a self-evolutionary strategy to uncover programs without the need for any ground truth data. Notably, these programs not only outperform state-of-the-art methods on average but also offer interpretability and exhibit robust transferability across diverse datasets and domains.

**Claims And Evidence:**

Yes, the evidence is supported by clear and convincing evidence

**Essential References Not Discussed:**

The author should have discussed most of the relevant works.

**Experimental Designs Or Analyses:**

The design of the experiment is comprehensive.

**Methods And Evaluation Criteria:**

Yes, the proposed method makes sense for the application.

**Other Comments Or Suggestions:**

Answer the W1-W2

**Other Strengths And Weaknesses:**

**Strengths**

S1. GRAIL is novel in that it uses a large language models generator to calculate graph editing distance (GED).

S2. The manuscript is well written, such as the Introduction Section provides an easy access to the existing challenges and contributions.

S3. The analysis of the experiment seems very reasonable and comprehensively shows that the proposed method is effective.

**Weaknesses**

W1. I am concerned that the effectiveness of this work is heavily reliant on the capabilities of large language models, rather than the inherent architectural design proposed in the paper.

W2. In the main text, there is a lack of detailed introduction of related works, which is not conducive to readers who are not familiar with the field to understand the background and significance of this work.

**Questions For Authors:**

Answer the W1-W2

**Relation To Broader Scientific Literature:**

The use of large language models will bring new insights and inspiration to the field.

**Theoretical Claims:**

There are many theoretical proofs in the appendix, and they all seem to be very solid.

---

> ### Author Rebuttal · Authors · 2025-03-29
>
> Thank you for the constructive feedback on our work. Below, we outline the changes made to address the reviewer's concerns. If the reviewer finds our responses satisfactory, we would sincerely appreciate a reconsideration of our paper’s rating.
>
> -------------
>
> **W1.** **I am concerned that the effectiveness of this work is heavily reliant on the capabilities of large language models, rather than the inherent architectural design proposed in the paper.**
>
> **Ans:** Our approach has four main components:
>  1. Reducing GED to bipartite matching
>  2. Learning weights of the bipartite graph using LLM
>  3. Prompt-tuning:
>      - Scoring programs using submodular optimization
>      - Genetic evolution.
>
> Fig 3c in the manuscript is an ablation study showing that submodular optimization indeed helps (lines 417--430).
>
> To further enhance our ablation study, we have conducted the following additional experiments to establish that just the LLM is not sufficient to obtain good performance.
>
> 1. **Is reduction to bipartite matching necessary?** We directly ask the LLM to predict code for GED instead of weights of the bipartite graph. The results deteriorate dramatically as shown below (Direct).
>
> |    | DATASET| AIDS  | Linux | IMDB  | ogbg-molhiv | ogbg-code2 | ogbg-molpcba |
> |-----------|-----------|-------|-------|-------|---|---|-----|
> | **RMSE**      |  **GRAIL**         |            **0.57**  | **0.13**  | **0.55**  | **2.96**        | **4.22**        | **3.18**  |
> |           |  **Direct fns (15)**              |7.95  | 2.81   | 112.64| 7.47        | 25.68       | 9.71        |
> | **EMR**|**GRAIL**              |**0.8**     |**~1**    | **~1**    | **0.2**         | **0.12**        | **0.12**        |
> |            |  **Direct fns (15)**              | 0.01  | 0.14  | 0.09  | 0.02        | 0           | 0.02        |
>
> 2. **Is genetic evolution necessary?** We randomly select programs from our pool for the prompt instead of genetic algorithm. The RMSE increases by 2 to 3 times on average.
>
> | Metric | Method  | AIDS  | LINUX | IMDB  | ogbg-molhiv | ogbg-code2 | ogbg-molpcba |
> |--------|--------|-------|-------|------|-------------|------------|--------------|
> | RMSE   | GRAIL  | **0.57** | **0.13** | **0.55** | **2.96**    | **4.22**   | **3.18**     |
> |        | Random | 0.94  | 0.38  | 0.93    | 3.48        | 4.42       | 4.19            |
> | EMR    |GRAIL  | **0.83** | **~1** | **0.99** | **0.18**    | **0.11**   | **0.12**     |
> |        | Random | 0.67  | 0.97  | 0.97    | 0.14        | 0.10       | 0.07           |
>
> **W2.** **In the main text, there is a lack of detailed introduction of related works, which is not conducive to readers who are not familiar with the field to understand the background and significance of this work.**
>
> **Ans:** We propose adding the following more detailed description of related works in the main manuscript. We will also add an even more detailed version, particularly on neural methods, in the appendix.
>
> >### **Non-Neural Methods**
> >Computing GED exactly or approximating it within provable bounds is challenging, leading to the development of various heuristic approaches \citepBlumenthal}. These methods utilize techniques such as transformations to the linear sum assignment (**NODE** \citep{NODE_ADJ_IP}, **BRANCH-TIGHT** \citep{BRANCH_TIGHT}), mixed integer programming (**MIP**) (**LP-GED-F2** \citep{lerouge2017new}, **ADJ-IP** \citep{NODE_ADJ_IP}, **COMPACT-MIP** \citep{COMPACT-MIP}), and local search methods (**IPFP** \citep{leordeanu2009integer}).
>
> >Unlike black-box neural methods, these approaches not only approximate GED but also provide the edit path, offering insights into structural modifications. However, their *approximation quality* is often inferior to neural approaches~\citep{ranjan2022greed, simgnn}, driving the shift toward neural architectures. Additionally, these methods often involve solving complex optimization problems, such as **MIP**, to derive node alignments between graphs.
>
> >### **Neural Methods**
> >Recent advancements favor graph neural networks (GNNs) for GED approximation due to their superior accuracy over non-neural methods \citep{ranjan2022greed,h2mn,simgnn,piao2023computing,genn,eric,graphedx,graphsim,graphotsim,icmlged}. These models take pairs of graphs with known GED values as input and are trained to predict GED distances. However, since computing true GED is NP-hard, training these models efficiently for large graphs or datasets remains a significant challenge.
>
> >Among the leading algorithms, **GREED** \citep{ranjan2022greed} employs siamese GNNs with an inductive bias to learn GED while preserving its metric properties. **H$^2$MN** \citep{h2mn} utilizes a hierarchical hypergraph matching network for graph similarity learning. Other state-of-the-art approaches, such as **GEDGNN** \citep{piao2023computing}, **ERIC** \citep{eric}, and **GraphEdX** \citep{graphedx}, further explore GNN-based architectures for GED prediction.

---

### Official Review · Reviewer_YCYA · 2025-04-05

**Overall Recommendation:** 1

**Summary:**

This paper addresses the problem of computing graph edit distance. In contrast to existing neural and non-neural methods, the authors propose an LLM-based approach, referred as GRAIL, by transforming the graph edit distance computation into two sub-problems - (1) Weight selection in a bipartite graph and (2) budget constrained map selection. GRAIL overcomes limitations of neural methods by eliminating reliance on NP-hard ground truth data, achieving superior accuracy, cross-domain generalization, and interpretability across diverse datasets.

**Claims And Evidence:**

No - The problem of graph edit distance is decomposed into two subproblems of 1) Weight selection in a bipartite graph and (2) budget constrained map selection. However, it is not proved / discussed how this decomposition is going to help solving the problem. Please check the detailed comment below.

**Essential References Not Discussed:**

NA

**Experimental Designs Or Analyses:**

Yes

**Methods And Evaluation Criteria:**

No - the fundamental comparison with directly using LLM to solve GED is missing.

**Other Comments Or Suggestions:**

I would like to suggest the authors to think more on the weight selection in a bipartite graph subproblem. They should also investigate LLM's capability on solving GED directly in a fair way.

**Other Strengths And Weaknesses:**

**Strengths**:

1. The paper proposes to use LLMs to compute graph edit distance - which itself is very interesting.

2. The paper is very well-written. The overall flow is quite interesting. I really like the way authors put definitions and problems to make everything formal but still easy-to-follow.

3. Some experiments, especially the ablation studies are insightful.


**Weaknesses**:

1. There is a fundamental problem in the proposed idea. The authors decompose the problem of graph edit distance (GED) into two subproblems:
  - Weight selection in a bipartite graph
  - Budget constrained map selection

The authors show some submodular property based guarantee of a greedy algorithm for budget constraint map selection problem. But unfortunately they did not show the relation of the optimal solution of the budget constraint map selection problem with the actual problem they are trying to solve - which is graph edit distance. For example, if I solve the budget constraint map selection problem optimally for a given D = {P_1 ,..., P_m}, will it be able to achieve any guarantee of the performance on GED?

"The weight of an edge (v1, v2) is set based on some policy, which should ideally reflect the probability of v1 being mapped to v2 in the optimal GED mapping" - I think this subproblem (referred as *weight selection in a bipartite graph*) is at least as hard as learning GED between two graphs G1 and G2 directly. So it is not at all clear what is the advantage of decomposing GED into two subproblems. The solution of Subproblem 1 (*Weight selection in a bipartite graph*) in the paper is quite superficial in nature. It does not also have any guarantee.

2. The paper keeps claiming that GRAIL minimizes an upper bound of GED. However, it is never proved or clarified if this upper bound has any approximation guarantee on the optimal solution or is it a trivial upper bound? If the individual programs are not good, a subset of them minimizing Eq. 4 will not help solving GED.

3. The superior performance of GRAIL in Tables 3 and 4 are NOT attributed properly. Can it be just because of the inherent reasoning capability of LLMs? One baseline should have been using the same LLM to compute GED directly with some smart prompting and good / diverse in-context examples.

4. In L230, what is the guarantee that J(A_greedy) converges?

5. Although GRAIL demonstrates generalization to larger graphs, the computational overhead of generating and evaluating programs for dense or massive graphs may still pose challenges, especially in real-time applications.

**Questions For Authors:**

See above.

**Relation To Broader Scientific Literature:**

This paper is related to the broader literature on the computation of graph edit distance which is quite fundamental in graph ML.

**Theoretical Claims:**

No - I have not formally checked the correctness of the theoretical claims. But they intuitively seem to be correct.

---

### Decision · Program_Chairs · 2025-05-01

**Decision:**

Accept (poster)

**Comment:**

This paper proposes GRAIL, a two-stage approach using large language models (LLMs) to generate heuristic programs for approximating graph edit distance (GED). GRAIL aims to address limitations of existing neural and non-neural methods and demonstrates strong empirical results.

**Reviewer Comments:**
- **Reviewer vS7D** recognized the novelty of leveraging LLMs for GED and the comprehensive experiments, but questioned if the improvements were simply due to ensembling multiple scoring functions rather than methodological breakthroughs. After rebuttal, was more positive but recommended additional qualitative comparisons to prior heuristics.
- **Reviewers isZf and cLnD** appreciated the empirical results and clarity, but both expressed concern that GRAIL’s effectiveness may mostly derive from LLM capabilities rather than algorithmic advancement. They also noted limited discussion of related work and baseline diversity, though these were partially addressed in the rebuttal.
- **Reviewer YCYA** had major theoretical concerns: they questioned the motivation for decomposing GED into two subproblems and pointed out that the proposed decomposition and upper-bound minimization lack clear theoretical justification or guarantees. YCYA also noted the absence (at original submission time) of fair baselines directly prompting LLMs for GED, and remained unconvinced even after the rebuttal.

**Author Response:**
The authors provided new experiments and clarifications in response to reviewer critiques. However, the fundamental issues regarding the theoretical grounding and the actual advantage of the two-stage decomposition over direct LLM approaches remain insufficiently addressed.

**Conclusion:**
While this submission has some strengths in empirical evaluation, taking into account the substantive unresolved theoretical concerns and considering the overall quality of other papers in my batch, I recommend weak accept of this submission.